# The mechanisms underlying the enhanced high-temperature properties of GRX-810

Timothy M. Smith [1] ✉, Christopher A. Kantzos[1], Bryan J. Harder[1], Andreas Bezold[2], Milan Heczko [2], Jiashi Miao [2], Gabriel Plummer [3], Mikhail I. Mendelev[3], Aaron C. Thompson[4], Bernadette J. Puleo[1], Austin J. Whitt[4], Andreas Stark [5], Steffen Neumeier [6], Timothy P. Gabb[1], John W. Lawson[3], Michael J. Mills[2] & Paul R. Gradl [7]

The demand for metal alloys that can perform at extreme temperatures above 1100 °C while remaining manufacturable has sparked renewed interest in printable oxide dispersion strengthened (ODS) alloys. Recently, NASA developed an ODS alloy designed for additive manufacturing, known as GRX-810, which has demonstrated exceptional tensile and creep performance at temperatures of 1093 °C and higher. In the present study, tensile tests of GRX-810 are conducted up to 1316 °C and creep tests are performed in both the horizontal and vertical orientations, relative to the build direction. Thermal cycling is executed at 1100 °C, 1200 °C, and 1300 °C in air. The oxidation behavior of GRX-810 is compared to that of alumina forming single crystal Ni-base superalloys and chromia-forming wrought alloys such as superalloys 718 and 625. High resolution atomic-scale characterization and atomistic modeling are employed to explain the exceptional high temperature properties observed in GRX-810, particularly in relation to the unique, finer trigonal yttrium oxides produced during the additive manufacturing process.

Early industry adoption of metal additive manufacturing (AM) generated excitement due to its ability to produce near-net shape components that were traditionally difficult to manufacture through conventional wrought or powder metallurgy processes[1,2]. The capability to manufacture parts with features that were impossible to consider before AM, in addition to much faster production rates, resulted in early industry adoption, most notably in the aerospace industry[3,4]. Currently, AM components are being used in both commercial aircraft engines as well as propulsion systems for space launch vehicles[5]. Still, many of these AM components are being produced with legacy alloys that were originally developed for considerably different manufacturing processes (cast, wrought, etc.)[6,7]. With respect to high-temperature alloys that are produced through AM, materials such as superalloys 718[8,9], 625[10], and 230[11] are being widely employed despite these compositions not being optimized for the AM process.

However, recently, new materials have been brought to market that were developed specifically for AM[12–16]. These alloys possess compositions that are better suited to the rapid solidification rates and thermal cycling associated with AM[17]. In addition, AM also presents new opportunities to develop unexplored classes of materials by combining unlike systems during the printing process[18,19]. Many studies have found recently that AM can be leveraged to produce dispersion-strengthened alloys[20–23]. An early AM alloy to receive commercial success was GRCop-42, a Cu-based dispersion-strengthened, high-conductivity alloy used for combustion chambers[24]. Detailed characterization of the alloy found that the dispersion of the strengthening intermetallic $Cr_2Nb$ Laves phase was significantly

[1]NASA Glenn Research Center, Cleveland, OH, USA. [2]Department of Materials Science and Engineering, The Ohio State University, Columbus, OH, USA. [3]NASA Ames Research Center, Moffett Field, Cleveland, CA, USA. [4]HX5 LLC, Fort Walton Beach, FL, USA. [5]Institute of Materials Physics, Helmholtz-Zentrum Hereon, Geesthacht, Germany. [6]Department of Materials Science and Engineering, Friedrich-Alexander-Universität Erlangen-Nürnberg, Erlangen, Germany. [7]Propulsion Department, NASA Marshall Space Flight Center, Huntsville, AL, USA. ✉e-mail: timothy.m.smith@nasa.gov

enhanced when the alloy was 3D printed compared to traditional wrought processes[24]. Al alloys typically considered incompatible with welding and casting have also been 3D printed by coating the powder with inoculants—such as $TiB_2$ nanoparticles[25]. Both aforementioned examples rely on creating a phase dispersion during the AM process to improve printability or mechanical properties of the AM alloy.

Many studies have been published showcasing the ability of AM to produce Ni- and Fe-based ODS alloys through many different techniques. Some have employed ball milling or mechanical alloying to incorporate oxides into the metal powder feedstock before printing the material[26–28]. In these studies, defects such as agglomerated oxides and porosity are common due to the morphology of the powder and non-optimal oxide distribution from the ball milling process. Other efforts explored in situ alloying to produce a dispersion of oxides by introducing oxygen during the additive process[23]. Lastly, Martin et al.[25] first reported coating metal feedstock with inoculants through chemical reactions as a way of incorporating ceramic nanoparticles during AM[29]. In 2019, NASA introduced a technique that coated metal powder with $Y_2O_3$ nano-particles by leveraging a high-energy mixing method called "resonant acoustic mixing"[30]. This process does not use any chemical reactions, binders, or solutions to coat the powder, which is otherwise unaffected by the mixing process. This removes many difficult-to-control variables associated with past attempts to coat metal powder.

Using this coating technique, combined with thermodynamic modeling, NASA developed an ODS alloy known as GRX-810[31]. This alloy exhibited high-temperature properties that far surpassed those of other commercially available, high-temperature alloys currently used in AM. The creep strength/life of GRX-810 at 1093 °C was found to be orders of magnitude better than AM-processed superalloys such as 718, 625, and 230. Most notable was the improved high-temperature properties GRX-810 exhibited over other ODS alloys in the same NiCoCr family, despite having seemingly similar compositions and microstructures[30]. One possibility for this difference could be the influence the additional reactive elements in GRX-810's nominal composition (Ni-32Co-30Cr-3W-1.5Re-0.8Nb-0.3Ti-0.3Al-0.055 C, in wt %) may have on the formation of nanoscale $Y_2O_3$ particles during the AM print process. Previous studies have demonstrated that $Y_2O_3$ will acquire other alloying constituents during processing (mechanical alloying, extrusion, etc.) to form modified nano oxides that deviate from a cubic crystal structure typical for $Y_2O_3$. Reactive elements such as Al[32], Ti[33], Hf[34], and Zr[35] have all been found to diffuse and react with $Y_2O_3$. For example, Al, in some instances, has been observed to react with $Y_2O_3$ to form monoclinic $Y_4Al_2O_9$[36]. These different oxides have been shown to possess varying particle size, contributing to differences in high-temperature strength[33].

In this study, we further investigate the high-temperature properties of GRX-810 (creep, tensile, oxidation) up to 1316 °C and explore how the alloy's microstructure influences these properties. Through atomic-scale microscopy and density functional theory (DFT) modeling, we provide insights into the dynamic processes that occur while manufacturing GRX-810 and how they explain the superior high-temperature properties found in GRX-810. Lastly, we compare the tensile strength and oxidation performance of GRX-810 to both conventional high-temperature AM alloys and single-crystal Ni-base superalloys such as B1900 (with and without Hf additions), SC-180, and CMSX-4. We show that GRX-810 exceeds the strength of conventional polycrystalline superalloys above 900 °C and single crystal superalloys above 1200 °C. GRX-810 also provides a favorable oxidation response compared to these alloys at temperatures exceeding 1300 °C. These exceptional results are explained by the presence of the first observed instances of trigonal (space group *P-3m1*) $Y_2O_3$ particles that exhibit significantly finer size distributions and greater number densities than the commonly found cubic $Y_2O_3$ particles characterized in other NiCoCr-based ODS alloys. These results highlight the significant impact that incorporating nano-scale oxides can have when combined with alloy compositions tailored for AM and high temperature environments.

## Results
### High temperature tensile results
In Supplementary Fig. 1a, the grain structure of GRX-810 produced using a 40 μm laser spot size (EOS M100) and an 80 μm laser spot size (EOS M280) is shown after being hot iso-statically pressed (HIP). The EBSD maps reveal a roughly 30% coarser grain structure in the material printed using a larger laser spot size. Though the grain structure is coarser, both materials exhibited a preferred [001] grain orientation and similar grain size along the build direction, as shown in a previous study[37]. For this study, material printed using the finer spot size will be considered fine grain (FG), while the other material will be labeled coarse grain (CG). In the build plane, these mean grain sizes are approximately 34 μm for FG and 45 μm for the CG. The larger grain size resulted in lower tensile strength at room temperature, presumably due to Hall-Petch effects. However, at elevated temperatures such as 1093 °C, the difference in strength caused by grain structure variability is significantly decreased[37]. In Fig. 1, the tensile strength of CG GRX-810 tested in both vertical and horizontal to the AM build direction is shown for room temperature, 1093 °C, and 1260/1316 °C conditions. For temperatures up to 1093 °C, the horizontal direction provided higher yield and tensile strengths, although at 1093 °C, the difference is much less pronounced. Notably, the elongation of the horizontal direction was comparable to the vertical direction at room temperature but is significantly reduced at 1093 °C (15% vs 55%). However, this elongation difference again converges at temperatures near melting, >1260 °C.

In Supplementary Table 1, all the elevated temperature tests for GRX-810 in as-built (non-heat treated) and HIP conditions for the

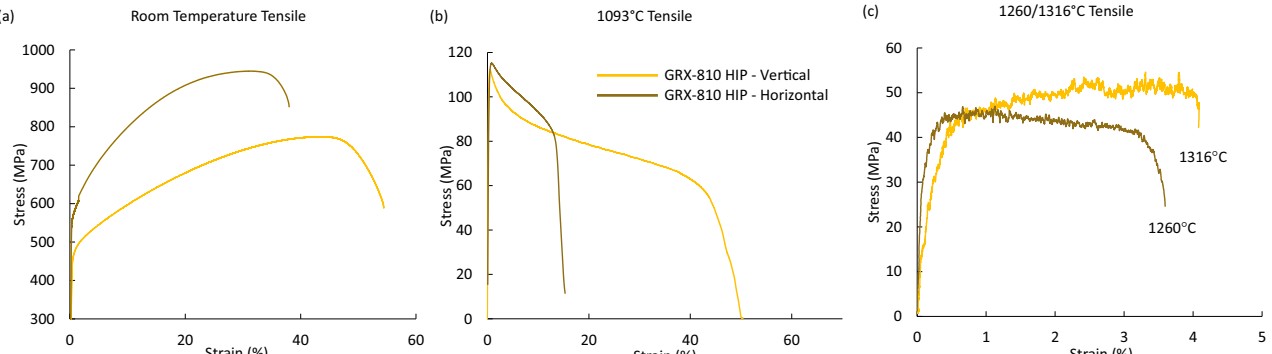

**Fig. 1 | Mechanical tests of CG vertical and horizontal GRX-810. a** Room temperature, **b** 1093 °C, and **c** extreme temperature (1260/1316 °C) tensile curves of HIP'ed GRX-810 in the vertical and horizontal directions. HIP Hot isostatic press.

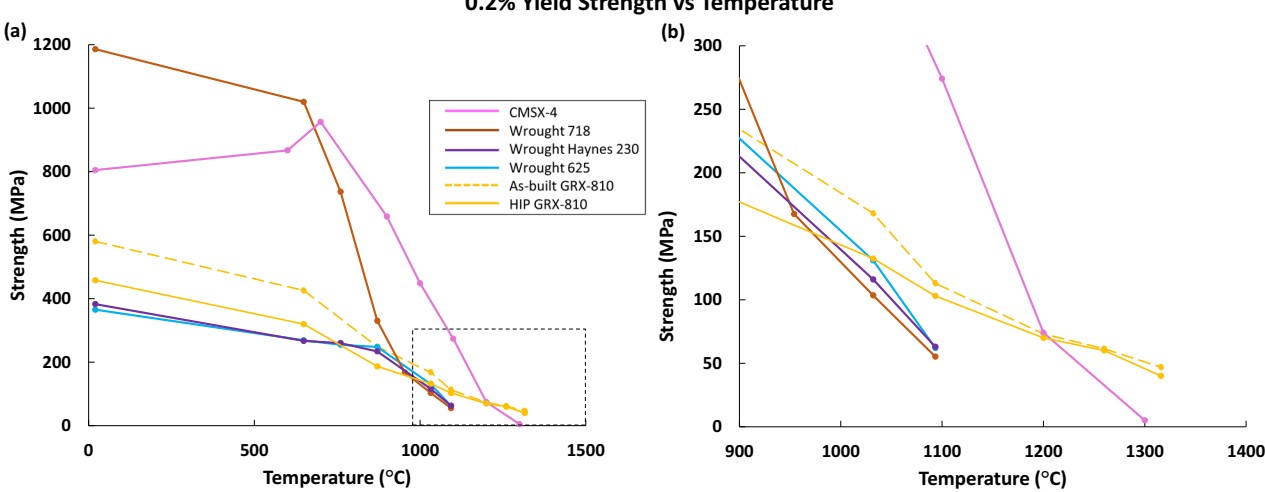

**Fig. 2 | Yield strength comparison of GRX-810 vs commercial superalloys[38,83].**
**a** Yield strength vs. temperature plots for the single crystal CMSX-4, wrought 718, wrought H230, wrought 625, vertical as-built CG GRX-810, and vertical HIP CG GRX-810. **b** The same curves highlighted in the dashed black box in (**a**). HIP Hot isostatic press.

vertical orientation and horizontal orientation are provided. A previous study indicated strength differences between CG and FG GRX-810[37]. The study also showed that HIP'ed GRX-810 possesses less strength compared to its as-built state, especially at lower temperatures. From the results in Supplementary Table 1, the effect of different grain structures and post-processing on tensile properties is minimized at temperature above 1093 °C. Another notable behavior is that GRX-810 still possesses significant strength at 1316 °C (>97% of the melt temperature [$T_m$] (solidus) of GRX-810 as revealed in Supplementary Fig. 1b). Figure 2 compares the CG GRX-810 strength in the as-built and HIP condition to the strengths of conventionally used superalloys (718, 625, H230, single crystal CMSX-4) as a function of temperature.

At room temperature, the strength of GRX-810 exceeds that of other solid solution strengthened wrought alloys such as 625 and H230; but it is significantly less than the precipitation strengthened alloys 718 and CMSX-4. However, by 950 °C and above, the strength of GRX-810 surpasses that of 718. At that temperature, it becomes evident that grain boundaries begin limiting the strength of metallic alloys when comparing the polycrystalline alloys to the single crystal CMSX-4. Despite possessing a relatively fine-grain structure, GRX-810 surpasses the single-crystal CMSX-4 in yield strength at 1200 °C and above[38]. This result is presumably a consequence of the strengthening precipitates in CMSX-4 dissolving at these temperatures, which severely reduces the strength of the single crystal[38]. Figure 2 also shows that the conventional (non-ODS) alloys have a point at which the strength drops precipitously, which marks a hard temperature limit for use. However, in GRX-810, no such limit was observed up to 1300 °C in both build orientations, which indicates this material may show a much more gradual drop in mechanical properties even at temperatures beyond those explored in this study.

**Vertical and horizontal creep strength**
The differences in tensile ductility and build direction highlighted in Fig. 1 at 1093 °C suggest that anisotropies could exist for other mechanical properties, such as creep. Previous studies have highlighted that creep can be strongly influenced by grain structure and size[39–41]. Therefore, it becomes important to better understand the effect that grain structure may have on AM ODS alloys such as GRX-810.

The creep curves highlighted in Fig. 3 reveal the differences in creep strength for GRX-810 when printed with a fine grain structure compared to a CG structure. Though the grain sizes are not vastly different, as shown in Supplementary Fig. 1, they result in large differences in creep life at low stress levels. At 21 MPa and 1093 °C, CG-GRX-810 reached 1% creep strain at around 5000 h while FG GRX-810 reached 1% creep strain in 2800 h. For comparison, AM 718 was reported to reach 1% creep strain under these conditions in 2.2 h[31]. At double the stress level, 41 MPa, the CG-GRX-810 material provided 1800 h of creep life compared to just 180 h in the FG sample. Interestingly, at 62 MPa both the FG and CG GRX-810 material failed at ~10 h, suggesting that the primary failure mechanism is less sensitive to grain structure at these higher stresses.

Conventional ODS alloys produced via directional recrystallization generate large, elongated grain structures that result in severe creep anisotropy[42]. Since AM-processed GRX-810 creates similar, though not as prominent, elongated grain structures along the build direction, creep tests were performed in the horizontal orientation to determine the differences in creep strength that may exist.

In Fig. 4, the horizontal creep curves for HIP CG GRX-810 are shown at 21, 31, 41, and 52 MPa compared to horizontal NiCoCr-ODS tested at 21 MPa. When comparing this data to the data in Fig. 3, the creep anisotropy that existed in conventional ODS alloys is clearly still present for ODS alloys that are 3D printed. However, despite the significant reductions in creep life for the horizontal orientation, GRX-810 still possesses high creep rupture ductility and life compared to conventional AM superalloys such as superalloys 718 or 625[31]. More notably, the horizontal creep performance of GRX-810 is orders of magnitude better than horizontal NiCoCr-ODS at 21 MPa, which ruptured before reaching 1% creep strain.

**Elevated temperature cyclic oxidation**
Historically, temperatures of 1300 °C are beyond what would be normally considered usable for chromia or alumina-forming alloys[43]. However, the high tensile strength exhibited by GRX-810 at 1260 °C and 1316 °C suggests that there may be applications at these temperatures where GRX-810 could be employed. Therefore, to better understand the oxidation performance and microstructural stability of as-built FG GRX-810, 1-h isothermal cycling was performed in air at 1100 °C, 1200 °C, and 1300 °C. The specific weight change results of these tests for GRX-810 are compared to other superalloy classes in Fig. 5.

At the three temperatures shown in Fig. 5, both single-crystal alumina formers (B-1900 and SC-180) and predominantly chromia

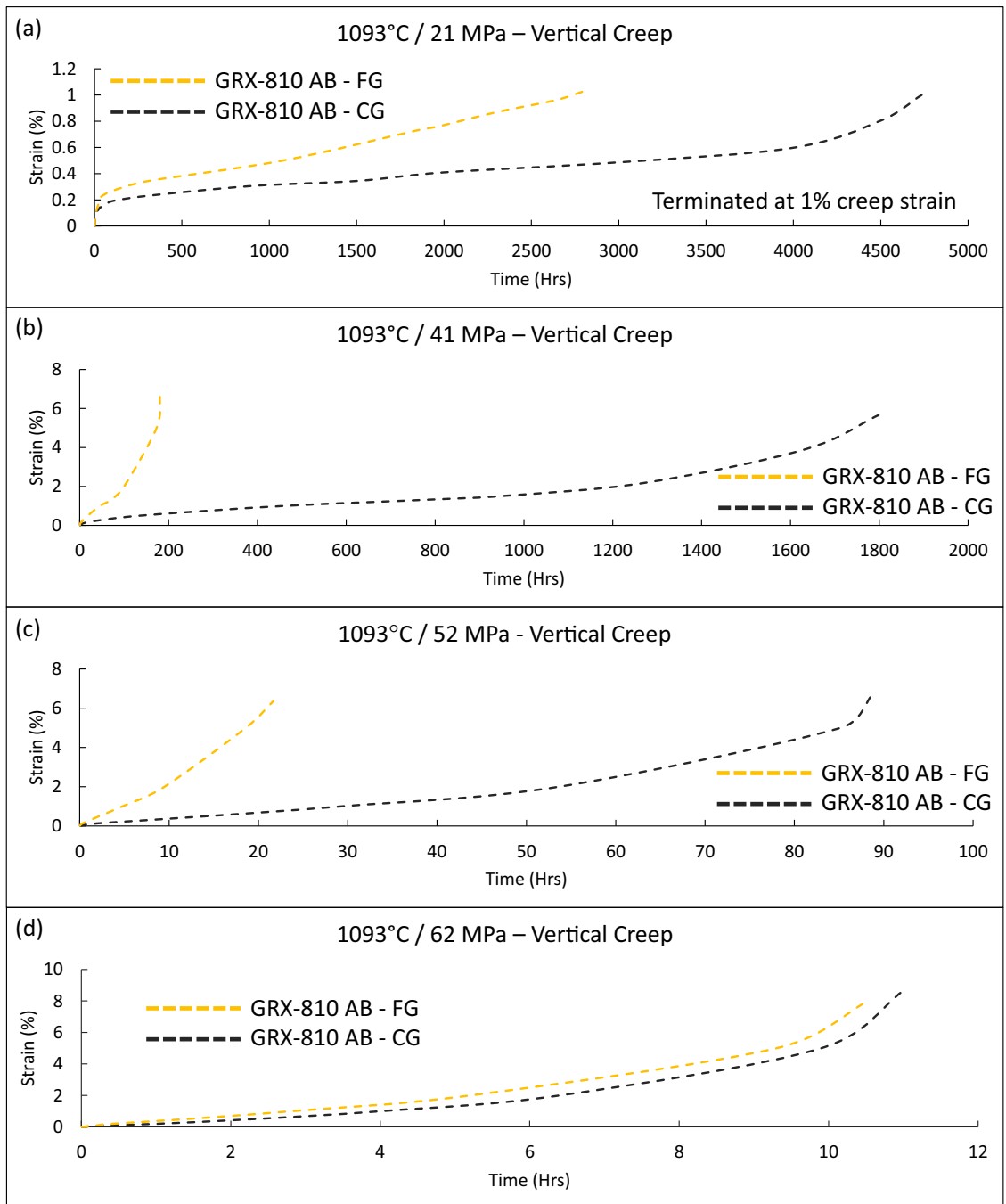

**Fig. 3 | Vertical creep curves of GRX-810 at elevated temperature.** Creep curves of vertical as-built fine grain (FG) and coarse grain (CG) GRX-810 at 1093 °C **a** 21 MPa, **b** 41 MPa, **c** 52 MPa, and **d** 62 MPa. The creep tests at 21 MPa (**a**) were terminated at 1% creep strain.

formers (GRX-810, ME3, 718, 625) were evaluated. In Fig. 5a, a significant difference in oxidation response is observed between an alumina protective oxide scale (B-1900) vs. a majority chromia protective scale (718). Despite forming a mixed oxide scale, the rate of specific weight change of GRX-810 was closer to B-1900 than Inconel 718, suggesting a higher temperature capability than traditional chromia-forming alloys, even though GRX-810 does not possess enough Al (0.3 wt%) to form an alumina scale[44]. At 1200 °C, a similar result was found and shown in Fig. 5b. Chromia-forming alloys, ME3 (a disk alloy) and 718 failed catastrophically (sample weight below 1 gram) at 60 and 80 h, respectively. Superalloys 625 and GRX-810 completed the full 200 1-h cycles and remained intact without distortion. Ultimately, both traditional and ODS chromia-forming alloys exhibited more weight

loss than the alumina-forming single crystal B-1900 alloy up to 1200 °C. This result is expected as alumina is known to provide better oxidation protection at higher temperatures than chromia, which becomes unstable at 1100 °C[45].

At 1300 °C, oxidation takes place much faster in all the tested alloys, as shown in Fig. 5c. Both alumina and chromia-forming alloys exhibit rapid oxidation, and in many cases, low-melting-temperature Ni and Co oxides appeared after 1 h of exposure. Superalloy 625, B-1900, and B-1900+Hf were all terminated after the first cycle due to observations of melting, high weight loss rates measured, or both. Superalloy 718 ran for three cycles until significant swelling and weight gain occurred, as shown by the arrow in Fig. 4c. After seven cycles, the 718 sample was removed for characterization, as shown in Fig. 6b.

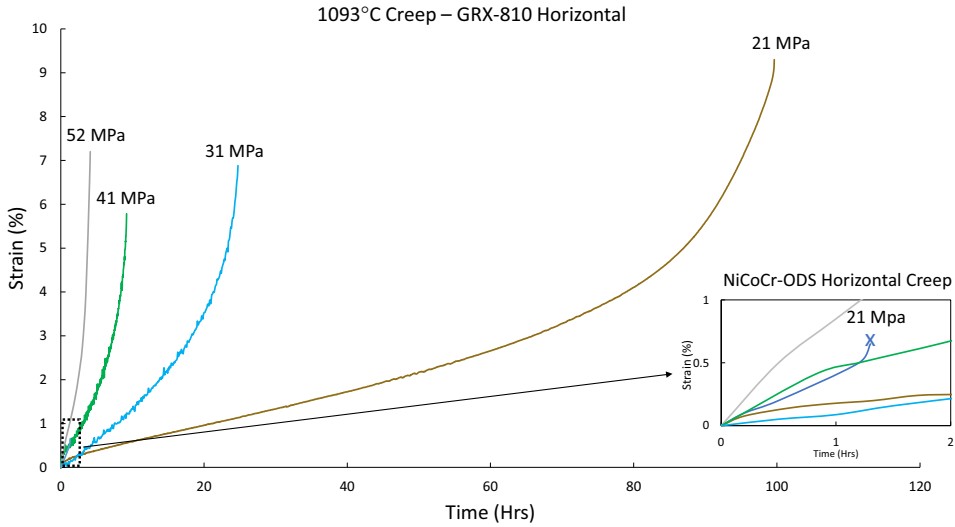

**Fig. 4 | Horizontal creep curves of GRX-810 at elevated temperature.** Creep curves of horizontal HIP CG GRX-810 at 1093 °C under various stresses. Lower right: same curves compared to horizontal NiCoCr-ODS at 21 MPa.

Regarding the alumina-forming alloys, SC-180 provided better performance at 1300 °C than B-1900, making it to 10 h. However, the weight loss rate steadily increased via shedding of large oxide phases (and observed melting at the coupon edges) with each subsequent exposure. Unexpectedly, GRX-810 performed the best at these extreme temperatures. The weight loss rate did not indicate runaway oxidation or impending catastrophic material loss. Instead, it reached a steady state weight change rate after 4 h, similar to what was observed at 1100 °C and 1200 °C via constant oxide formation at high temperature and spallation upon cooling to room temperature. The samples maintained their integrity as well, showing minimal distortion after 10 h of exposure (inset in Fig. 5c), although there was observable thinning of the sample. Therefore, these results indicate that ODS-chromia forming alloys outperform conventional chromia forming alloys from 1100 to 1300 °C, as well as some alumina forming alloys at temperatures 1300 °C and above. This trend is similar to that observed for the high-temperature tensile tests.

To better understand the oxidation results, an additional 1300 °C cyclic oxidation tests were performed comparing GRX-810 and the current state-of-the-art single crystal blade alloys CMSX-4 and CMSX-10, as shown in Supplementary Fig. 2. The results in Supplementary Fig. 2 appear to further confirm the high temperature oxidation test shown in Fig. 5c. GRX-810 had a slower weight loss rate compared to CMSX-10. CMSX-4 had the best oxidation performance over the 20 cycles measured; however, after 15 cycles its weight loss rate started to accelerate and exceed that of GRX-810. This change may be caused by a depletion of Al near the surface, compromising the alloy's ability to maintain a stable alumina oxide layer. Experiments performed by Wang et al. on the sublimation of $Cr_2O_3$ at high temperatures projected ~$5 \times 10^{-8}$ gm/$cm^2$ sec[46]. The weight loss observed in Fig. 5c was on the order of ~$2 \times 10^{-4}$ gm/$cm^2$ sec, so it is assumed that the weight change was dominated by oxide scale spallation and not volatilization of chromia at elevated temperatures. Additionally, a 24-h soak at 1300 °C was performed on GRX-810 caused significant weight gain, which further supports the hypothesis that the weight loss in cyclic testing is from spallation and not sublimation. Figures 6a, b indicate significant differences in microstructural stability between 718 and GRX-810 when cycled at 1300 °C.

Figures 6a, b indicate significant differences in microstructural stability between 718 and GRX-810 when cycled at 1300 °C. After 10 cycles, GRX-810 has an oxide layer on the surface of the sample that is 10–20 μm thick, while the 718 sample was oxidized completely through the bulk of the material. 718 also exhibited large porosity and grain growth, as shown in Fig. 6a. This porosity is most likely caused by incipient melting or rapid diffusion of oxidant and reaction with alloying elements, such as Cr[47]. No such porosity or incipient melting was observed in any of the GRX-810 samples. SEM analysis of the oxide layer formed in GRX-810 at these conditions reveals a continuous $Cr_2O_3$ oxide layer with $Y_2O_3$, as shown in Fig. 6c. The observation of a continuous oxide layer, void formation, and Cr depletion near the surface of the sample suggests the oxidation performance of GRX-810 is driven more by spallation than vaporization. In Supplementary Fig. 3a–c, the oxide morphology and size distribution between as-built GRX-810 and after ten 1-h cycles at 1300 °C are shown through SEM images. Additionally, Supplementary Fig. 2b reveals that no apparent oxide coarsening or deleterious phases are observed after 10 h at 1300 °C, which indicates both the GRX-810 compositional stability as well as oxide stability even at temperatures near melting (measured solidus is 1357 °C as shown in Supplementary Fig. 1)[31]. The grain orientation map in Supplementary Fig. 3c reveals recrystallization of the grain structure after 10 thermal cycles at this temperature. Studies in the literature on ODS alloys indicate that the presence of nano-scale $Y_2O_3$ particles favors the growth of $Cr_2O_3$ over mixed oxide scales[48]. Thus, it is possible that the presence of nano yttria particles in the scale may also assist in adhesion, thus enhancing the durability of the alloy in these elevated temperature environments[49].

## Discussion

### Nano oxide characterization

The high-temperature mechanical properties and microstructural stability of GRX-810 clearly benefit from the incorporation of the nano-oxide particles into the solid solution matrix, similar to conventional ODS alloys produced through mechanical alloying processes. However, at present, little is understood about the structure, size distribution, and characteristics of these oxides and how they are incorporated into the matrix during 3D printing. Furthermore, notable differences have been observed between ODS alloys that were similarly mixed and printed, namely GRX-810 with superior mechanical performance, and NiCoCr-ODS (with and without Re and B additions (ReB)) with less impressive performance[31]. Thus, the oxides in NiCoCr-ODS (ReB) and GRX-810 were analyzed using high-resolution Scanning Transmission Electron Microscopy (STEM) imaging, X-ray energy dispersive spectroscopy (EDS), synchrotron diffraction, and atomistic modeling.

In Fig. 7a, b, a representative STEM micrograph reveals the atomic crystal structure of oxides found in GRX-810. Surprisingly, the

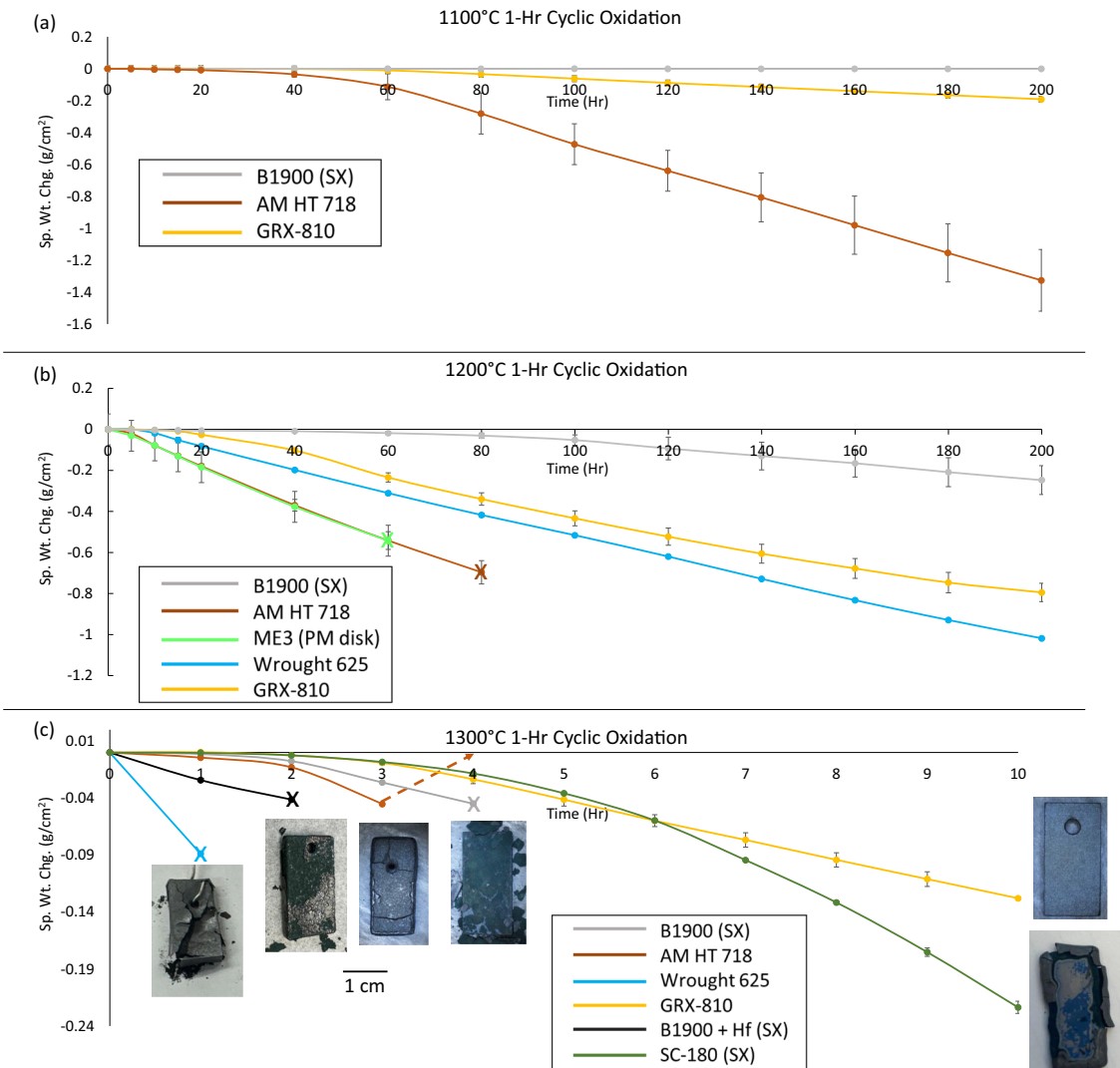

**Fig. 5 | Cyclic oxidation results at 1100 °C, 1200 °C, and 1300 °C.** Cyclic oxidation results for **a** GRX-810, 718, and single crystal B1900 at 1100 °C, **b** GRX-810, 718, ME3 (powder metallurgy (PM) disk alloy), 625, and B1900 at 1200 °C, and **c** GRX-810, 625, 718, single crystals B1900+Hf, B1900, and SC180 at 1300 °C. Images of the samples were taken after the final cycle measured to highlight the state of the alloy at that point. Note: the 718-sample produced runaway oxidation gain after cycle 3, which has been denoted with the dashed arrow. "x" denotes a test ending due to catastrophic oxidation (sample weight <1 g or severe microstructure degradation). SX = single crystal. Error bars represent 1 s.d. using a minimum of three measurements.

crystal structure of the $Y_2O_3$ in GRX-810 has transformed into a trigonal structure compared to the expected cubic structure found in the NiCoCr-ODS (ReB) alloy, as shown in Supplementary Fig. 4[50]. Figure 7b overlays the $Y_2O_3$ trigonal crystal structure with the atomic resolution STEM image showing a convincing match, which has also been verified with additional imaging along several other zone axes (see Supplementary Fig. 5). A fast-Fourier transformation of the image in Fig. 7b is shown in Fig. 7c, supporting that the trigonal crystal structure is present. This is an unexpected result considering the X-ray diffraction analysis shown in Supplementary Fig. 6 confirmed that the $Y_2O_3$ nano oxide powder is cubic before the printing process. The trigonal structure of the oxides in GRX-810 was confirmed using synchrotron-based high-energy X-ray diffraction (HEXRD), as shown in Fig. 7d. The oxides in NiCoCr-ReB ODS were again found to be cubic. A small number of $Y_4Al_2O_9$ oxides were also found in the NiCoCr-ReB ODS specimen, but were excluded from the analysis as their presence could be attributed to contamination during the AM print process. The presence of trigonal $Y_2O_3$ is particularly unexpected as it has never been reported before in ODS alloys. Indeed, the trigonal crystal structure has never been observed

at ambient temperature and pressure in bulk before[51]. Bulk $Y_2O_3$ has three crystal structure variants that are discussed in the literature at ambient pressures. The most common is the C-type $Y_2O_3$, which possesses a cubic crystal structure (space group *la-3* no. 206)[52,53]. $Y_2O_3$ will undergo a crystallographic transformation right before melting at 2347°C to the H-type hexagonal (space group *P63/mmc*) phase. However, the H-phase cannot be stabilized by quenching, and $Y_2O_3$ always reverts to C-type upon cooling[53]. The other commonly observed $Y_2O_3$ phase observed is the B-type monoclinic $Y_2O_3$ (space group *C2/m*). B-type $Y_2O_3$ becomes stable under high pressures (5-7 GPa). $Y_2O_3$ will maintain the metastable B-type crystal structure when the pressure is released. This phase transformation can also occur through doping $Y_2O_3$ with other elements. If $Y_2O_3$ is placed under more extreme pressures (>10 GPa), then another phase transformation will occur to a trigonal A-type (space group *P-3m1*)[52]. However, this structure is unstable when the pressure is released, and the bulk $Y_2O_3$ will revert to the metastable B-type phase. Therefore, A-type $Y_2O_3$ has never been observed to be stable at ambient temperature or pressure, thus, the observation of trigonal oxides in GRX-810 must be scrutinized to ensure a correct conclusion.

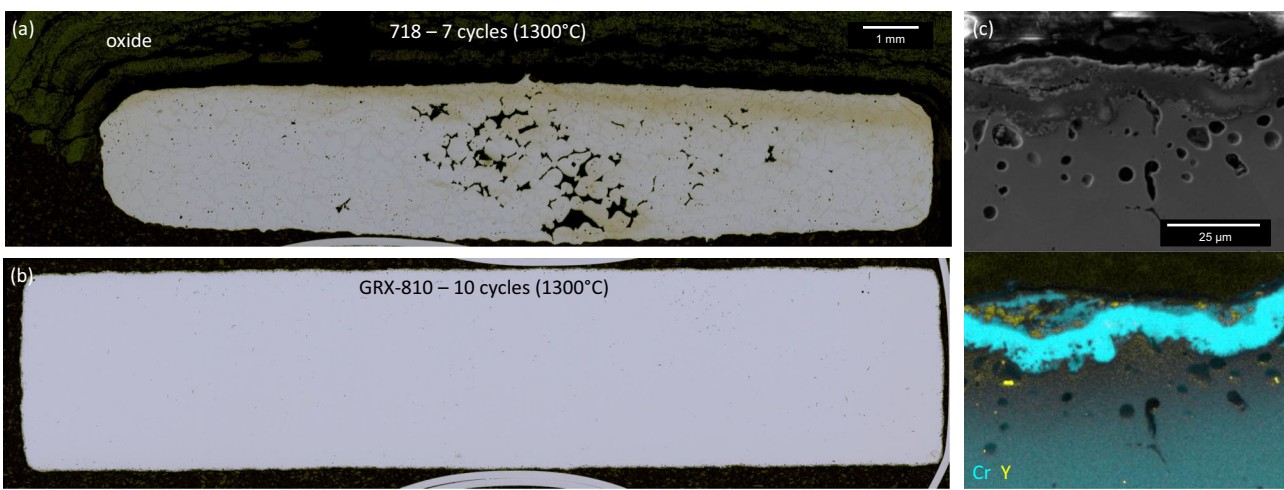

**Fig. 6 | Oxidation and microstructural stability at 1300 °C.** Metallography cross-sections of **a** superalloy 718 after 7 cycles and **b** GRX-810 after 10 cycles. **c** SEM and chemical map revealing a continuous $Cr_2O_3$ oxide layer with $Y_2O_3$ in the GRX-810 oxide scale from (**b**).

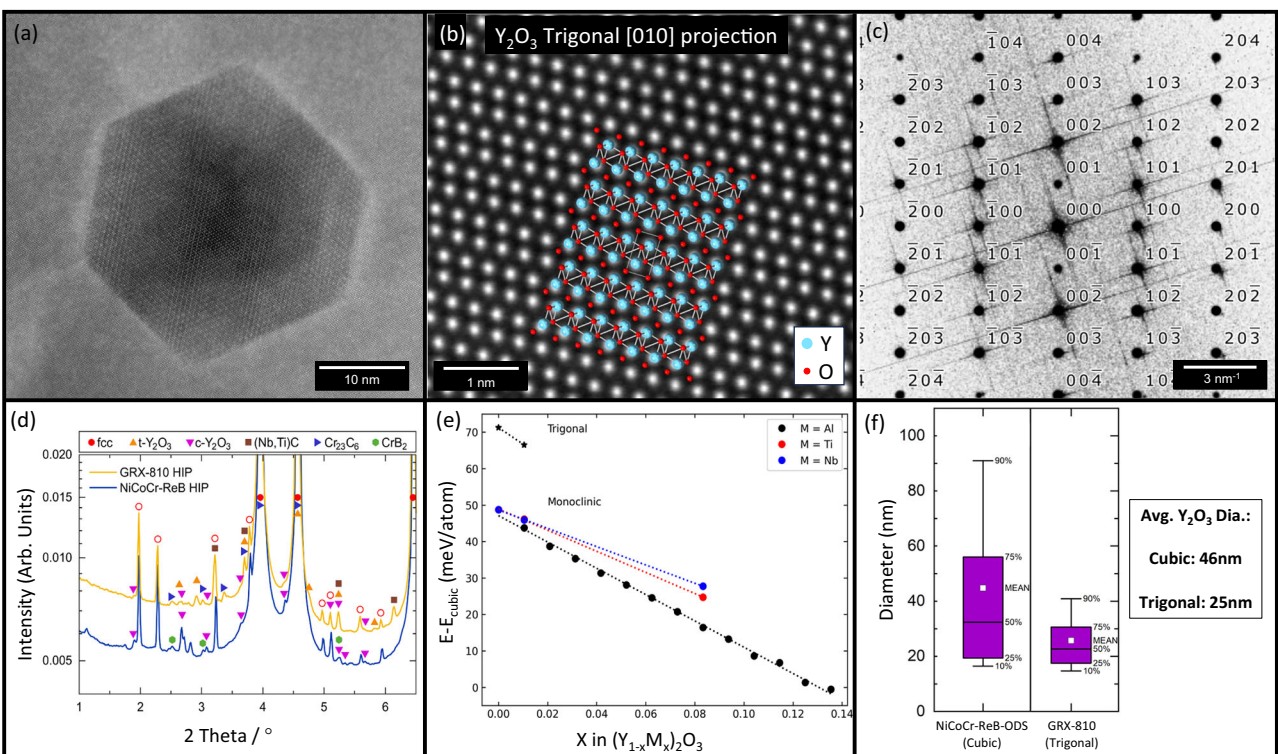

**Fig. 7 | High-resolution characterization of oxide nanoparticles.** Atomic-scale STEM micrographs of oxides found in GRX-810 **a** revealing faceted oxide faces and **b** trigonal crystal structure. **c** The fast Fourier transformation from **b** supporting the trigonal crystal structure. **d** A comparison of the oxides in NiCoCr-ReB and GRX-810 using synchrotron-based high-energy X-ray diffraction confirms the presence of trigonal oxides in GRX-810 and cubic oxides in NiCoCr-ReB ODS. **e** DFT calculations for the energy difference between cubic, monoclinic, and trigonal phases of $Y_2O_3$ as a function of substitutional defect concentration. **f** The size distribution of oxides in NiCoCr-ReB ODS and GRX-810 using high-resolution scanning transmission electron microscopy. The fit curves represent the best-fit linear equation.

Figure 7 already reveals multiple characterization tools suggesting a trigonal crystal structure, but the similarity between the H-type hexagonal and A-type trigonal structures makes it difficult to distinguish between these structures using diffraction techniques. Supplementary Fig. 5 reveals the fast Fourier transform (FFT) maps pertaining to high-angle annular dark-field (HAADF) images of the oxides in GRX-810 along several zone axes, and the expected patterns for the hexagonal and trigonal structures. In all cases, the FFT maps confirm the earlier stated observation of stable trigonal $Y_2O_3$ oxide nanoparticles in an ODS alloy. Atomistic modeling was performed to better elucidate the effect that Al, Ti, and Nb segregation may have on the stability of the different crystal structure types for $Y_2O_3$. From the calculation shown in Fig. 7e, monoclinic $Y_2O_3$ becomes the 0 K ground state structure at 13.0% Al occupation of the cation sites in $Y_2O_3$. The Ti and Nb substitutional defects had a similar but weaker effect than Al in making the monoclinic phase of $Y_2O_3$ more favorable as their concentrations increased. A similar trend can be observed for the trigonal crystal structure. However, STEM-EDS did not reveal a prominent segregation of these elements in the oxides that could fully explain the presence of the type-A oxides in GRX-810. Supplementary Fig. 7 does reveal carbon segregation near the oxide/matrix interface and suggests that the effect carbon has on $Y_2O_3$ phase stability should be investigated in future studies.

From Fig. 7 and Supplementary Fig. 5, it can be concluded that the oxides in GRX-810 are in fact Type A, trigonal $Y_2O_3$ nanoparticles. Considering that both NiCoCr-ReB ODS and GRX-810 were manufactured using the same process and the same initial $Y_2O_3$ nanoparticles, the difference in oxide crystal structures between the two alloys suggests that alloy composition, in combination with the rapid solidification of AM, can promote the appearance of Type-A trigonal $Y_2O_3$ nanoparticles. While fully describing the formation of these unique oxides will be the scope of future studies, some initial considerations are provided here. Ab initio molecular dynamics (AIMD) simulations reveal that the trigonal phase is mechanically stable only at very high (close to melting) temperatures. Thus, its appearance in combination with the small sizes of the $Y_2O_3$ particles in GRX-810 is most likely associated with the melting and precipitation of $Y_2O_3$ during AM[54]. This is broadly consistent with experiments, which show a phase transition from the cubic phase to the hexagonal H-type phase around 100 °C below the melting temperature[55]. The reactive elements in GRX-810 may help promote both the H-type and/or A type phases, as is suggested in Fig. 7e. The promotion and rapid quenching of the H-type phase could result in formation of the type-A trigonal phase, which then remains due to the rapid solidification of the metal surrounding the oxide as a consequence of the large difference between the trigonal and cubic phase, as shown in supplementary Fig. 8. This may explain why the trigonal phase is promoted and observed in the GRX-810 at the room temperature though it is not stable in bulk at these conditions.

## Oxide dispersion strengthening in GRX-810

It remains an open question as to whether this difference in oxide crystal structure can explain some of the high-temperature performance differences between the two ODS alloys. In a previous study, Zhou et al.[56] explored the compositional effect on oxide formation and characteristics for Fe-Cr-Al based ODS alloys. They found that different oxide particles and their crystal structures were correlated with various size distributions of the oxides. In their study, they found monoclinic $Y_4Al_2O_9$ was associated with the finest size distribution. To investigate if a size distribution difference existed between the NiCoCr-ODS (ReB) and GRX-810 alloys, oxide size distributions were extracted from STEM-EDS Yttrium chemical maps and images, as shown in Fig. 7f. Similar to the finding by Zhou et al.[56], it appears that the trigonal oxides in GRX-810 are significantly finer than the cubic oxides in NiCoCr-ReB. Notably, 90% of the oxides in GRX-810 possessed a diameter less than the average cubic oxides in the NiCoCr-ReB ODS alloy. In fact, the average diameter (25 nm) of the trigonal oxides in GRX-810 was almost half compared to the average diameter of the cubic oxides found in the NiCoCr-ODS alloy (46 nm). This difference, though appearing subtle, suggests significant differences in the oxide size distribution, number density, and particle spacings between the two alloys. Considering the same oxide wt% was used for both alloys, a volume fraction of oxides can be estimated for each alloy based on the alloy density and the density of the different oxide types. Assuming a uniform distribution of spherical particles, a normal size distribution and using the measured average oxide diameter between the two alloys and calculated volume fractions, the average face-to-face distance between oxide particles ($\lambda$) can be determined using the relationship below[57]:

$$\lambda = 1.25 \left( \sqrt[3]{1/N} \right) - 2r \qquad (1)$$

where $r$ is the average radius of the oxide particle. The number density ($N$) can be determined with the relationship.

$$N = \frac{f}{V_{avg}} \qquad (2)$$

where $f$ is the volume fraction of the oxide particles and $V_{avg}$ is the average volume of an oxide particle. Using these relationships, the

average distance between the particles in GRX-810 and NiCoCr-ReB ODS was determined to be 70 nm and 124 nm, respectively. Notable is the difference in the calculated number density of oxides between the alloys. Assuming similar estimated volume fractions between the two alloys, GRX-810 was calculated to have a number density over five times greater than NiCoCr-ReB ODS ($1.88 \times 10^{21} m^3$ compared to $3.25 \times 10^{20} m^3$). Based on the significant oxide differences calculated between the two alloys, it should be expected that the high-temperature properties between the two alloys will be different. A calculated threshold stress at 1100°C for both alloys can be determined using the relationship below.

$$\sigma_{th} = A \left( \frac{M}{2\pi} \right) \left( \frac{Gb}{\lambda} \right) * \left( rmln \left( \frac{D}{r} \right) + B \right) \qquad (3)$$

where $G$ is the shear modulus of GRX-810 and NiCoCr-ReB ODS at 1100 °C, which was derived from $E$ and $v$ (0.26). $M$ is the Taylor factor (3.0), $A$ and $B$ are constants based on the dislocation characteristics and strengthening mechanisms and are 1.06 and 0.65, respectively. Lastly, $D$ is calculated from the relationship:

$$D = (2r\lambda)/(2r + \lambda) \qquad (4)$$

Based on these assumptions and relationships, the threshold stress for GRX-810 at 1100 °C was calculated to be 32 MPa while for NiCoCr-ReB ODS it was less than half that at 15 MPa. Though this calculation ignores much of the nuance and microstructure effects that control creep in these alloys (i.e., grain size, grain structure, remnant dislocation density, and grain boundary carbides), it clearly reveals the critical influence that particle size can have on the high-temperature creep properties of these alloys. The finer Type A trigonal oxides in GRX-810 can account for a 2× increase in the threshold stress compared to the larger type C cubic oxides found in NiCoCr-ReB ODS. As mentioned previously, earlier studies have explored producing a finer dispersion of oxides in ODS alloys through changes in the oxide chemistry and crystal structure. Most notable, Ti has been shown to promote $Y_2Ti_2O_7$ oxides that form much finer dispersions compared to cubic $Y_2O_3$[33]. However, this finer dispersion has been found to be a result of a greater coherency between these oxides and the surrounding metal matrix, which lowers the driving force, thus limiting growth of the oxide[58]. This higher coherency may be detrimental to the oxide's ability to pin dislocations, and may promote shearing of the oxides rather than climb-bypass[59]. GRX-810, manufactured using L-PBF, may be unique in that it has promoted a finer dispersion of $Y_2O_3$ without promoting oxides more coherent with the surrounding matrix, thus maximizing the strengthening potential of the oxide dispersion.

An additional case study was performed for this manuscript to quantify the important contribution that the trigonal $Y_2O_3$ oxides have on the high-temperature creep properties of GRX-810. Two different GRX-810 creep samples, one without the oxides and one with them, were manufactured using the same base powder feedstock, AM printer, and post-processing HIP step. The non-ODS GRX-810 sample was crept at 1093 °C/21 MPa, below the threshold stress calculated previously for ODS GRX-810. Both non-ODS and ODS versions were crept under the same conditions for as long as the non-ODS test survived. After the non-ODS GRX-810 sample ruptured, the ODS GRX-810 test was terminated, and both samples were characterized. The results are shown below in Fig. 8.

The creep curves reveal vastly different responses between ODS and non-ODS GRX-810 at 1093 °C/21 MPa. The non-ODS sample ruptured after 16 h at 4% strain, while the ODS sample did not creep at all. In fact, the ODS sample appeared to exhibit creep negative during the first few hours of the test. It is not clear if this is a true response or an artifact from the creep frame being revealed due to the lack of strain occurring during the test. Future investigations will further probe this

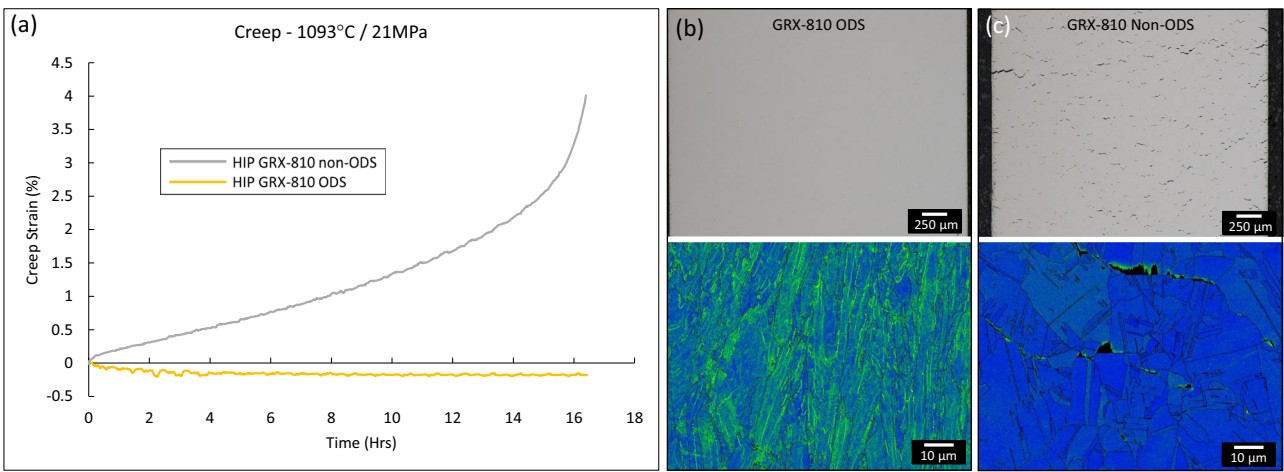

**Fig. 8 | Creep response between ODS and non-ODS GRX-810. a** The creep curves of non-ODS and ODS GRX-810. The ODS GRX-810 test was terminated after the non-ODS test failed at 16 h. The deformation and Kernel Average Misorientation maps of **b** ODS GRX-810 and **c** non-ODS GRX-810.

result. Still, the lack of a creep response in the ODS sample further validates the threshold stress calculated in this study. The analysis of the post-creep samples highlights the influence that the oxides play in the microstructural stability of GRX-810. In Fig. 8b, the ODS material did not reveal any deformation, recrystallization, or relaxation after both the HIP and creep test. The kernel average misorientation (KAM) map indicates that the dislocation substructure present after the AM printing process still remains after creep. In contrast, the non-ODS sample revealed a microstructure that fully recrystallized and possessed significant crack formation along grain boundaries. The KAM analysis of the non-ODS material revealed heavy deformation build-up at the grain boundaries and subsequent cracks, and an absence of the dislocation sub-structure that was initially present after the AM print. This comparison unequivocally confirms the importance these nano-oxides have on the exceptionally high-temperature properties of GRX-810.

### Orientation effects on GRX-810 creep properties

While the previous section explores the influence that the trigonal oxides have on the high temperature performance in GRX-810, the creep curves provided in Figs. 3 and 4 reveal that grain structure also plays an important role. An overview of these results, with additional creep tests, are highlighted in Fig. 9.

The Larson Miller Parameters revealed in Fig. 9a were calculated using the equation below:

$$Rupture\ LMP = (Temp(C^\circ) + 273.15)*(\log t_{hr} + 20) \tag{5}$$

Clear differences in creep rupture life of GRX-810 were observed between the vertical and horizontal build orientations. When designing parts for GRX-810, this creep anisotropy should be considered for complex stress states, however, preliminary component demonstrations have not shown unexpectedly poor performance for GRX-810[37]. Using the same creep tests, the creep stress exponents were calculated from the slope of the minimum strain rate as a function of stress for both orientations as shown in Fig. 9b using the standard relationship[60].

$$n = d(\ln(\dot{\varepsilon}))/d(ln\sigma) \tag{6}$$

The stress exponents reveal that the two orientations may be under different creep deformation failure modes. Higher values suggest dislocation creep is dominant, while lower values suggest that diffusion creep and grain boundary sliding are more active. The results in Fig. 9b suggest that grain boundary sliding mechanisms may be a more

prominent contribution in horizontally oriented creep compared to the vertical orientation in GRX-810. This result is supported by the fracture surfaces characterized in Supplementary Fig. 9. The fracture surface found in the horizontal GRX-810 creep specimen reveals that failure occurs primarily along grain boundaries. In contrast, the vertical GRX-810 creep specimen did not clearly appear to fail along grain boundaries.

It has been known for decades that ODS alloys exhibit creep anisotropy, where the horizontal orientation (transverse to the extrusion axis) produces similar creep debits[42,61,62]. A study by Totemeier and Lillo found that this anisotropy existed at temperatures of 1000 °C or higher[63]. At these elevated temperatures, creep failures in the transverse/horizontal specimens all failed by grain boundary separation. In general, any Ni/Co/Fe-based polycrystalline alloy with grain boundaries orthogonal to the loading direction will suffer a creep life debit[64]. Although not as severe, single-crystal blade alloys also exhibit significant creep differences based on crystallographic load orientation at these elevated temperatures[65,66]. One important aspect that differentiates GRX-810 from other conventional ODS alloys is that it is not plagued by the poor creep ductility (<1%) found in MA-754[63] and NiCoCr-ODS, as shown in Fig. 4. Though the creep strain rates are much faster in the horizontal direction, the creep ductility is surprisingly improved compared to the vertical creep curves (Fig. 3). Historically, the low transverse creep ductility of ODS alloys and property debits compared to the extruded axis[63] have negatively impacted industry adoption, even though the creep life and strain rates were usually improved compared to conventional alloys. It is presently unclear why GRX-810 circumvents this issue. One possibility is that the specific columnar grain structure[67] from the AM process is anisotropic enough to provide superior creep performance in the vertical build direction, but not sufficiently anisotropic to result in the creep ductility issues in the horizontal direction. Another study was performed comparing the rapid rupture creep of GRX-810 with and without the addition of oxides in the horizontal loading direction. The results are shown in Supplementary Fig. 10. The results reveal that, even in the inferior horizontal orientation, the fine trigonal oxides provide large improvements in creep life (two orders of magnitude longer) when compared to GRX-810 printed using powder without the oxide loading. Interestingly, the GRX-810 material tested in Supplementary Fig. 10 possessed a fine grain microstructure (due to being manufactured on the smaller M100 AM printer) and appears to perform better in creep at lower stresses than does the CG GRX-810 material shown in Fig. 4. Thus, the weaker creep life of GRX-810 in the horizontal orientation is a direct result of the detrimental grain structure

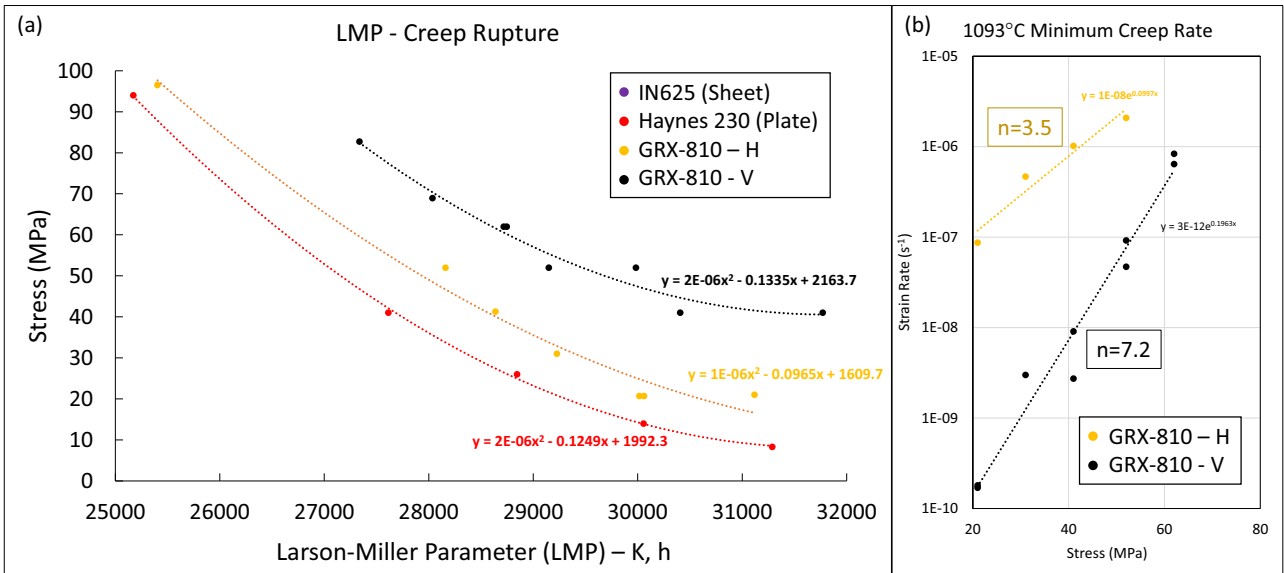

Fig. 9 | **Creep strength overview of GRX-810. a** The Larson Miller Parameter vs. stress for creep rupture in GRX-810 in the horizontal and vertical build directions compared to plate Haynes 230 and sheet 625 based on Eq. 5. **b** Creep strain rate vs. stress at 1093 °C for both orientations and the calculated stress exponents. H horizontal and V vertical. The fit curves represent the best fit **a** polynomial and **b** exponential equation for each data set.

with relation to the stress axis and not due to the addition of the nano-oxides. This is further confirmed by the lower creep life possessed by HIPed non-ODS GRX-810 at 20 MPa in the vertical direction compared to horizontal GRX-810, also shown in Fig. 8. In fact, any as-built AM superalloy would be expected to possess similar debits in creep in the horizontal orientation if they possess similar grain texture[68].

From Supplementary Fig. 10 and Fig. 8, it's clear that the oxides significantly improve the creep performance of GRX-810 in all orientations and are a major reason for the exceptional creep strength showcased by the alloy, especially in the vertical orientation. Still, are there other aspects of the alloy that could be further contributing to its exceptionally high-temperature strength? In the first publication of GRX-810[31], it was revealed that GRX-810 possessed thermally stable Nb/Ti-rich carbides on the grain boundaries. It is clear from Supplementary Fig. 9 that grain boundaries, especially those oriented orthogonal to the load axis, will be sources of failure during creep. These carbides may help in mitigating grain boundary sliding and thus improve creep. Beyond strengthening the grain boundaries, HR-STEM analysis of a post-creep GRX-810 samples revealed numerous nanoscale Nb and Ti-rich carbides precipitating in the bulk of material, further inhibiting dislocation motion and enhancing the dispersion strengthening of the alloy (See Supplementary Fig. 11). It can be assumed that the additional dispersion of nano-carbides further increases the already elevated threshold stress promoted by the finer trigonal oxides in GRX-810. Thus, the creep-induced carbides in GRX-810 may contribute to the alloy's high temperature properties by further inhibiting both grain boundary sliding and dislocation motion during creep in combination with the nano-oxides. Detailed characterization of the nature of these carbides, and their size and density, is currently in progress and will be reported elsewhere.

In this study, extensive testing and characterization were performed to better understand why GRX-810 has significantly improved high-temperature properties (>1100 °C) compared to other AM and AM-ODS alloys. Creep tests revealed that the grain structure produced during the AM process has an influence on the creep strength and strain rates for the alloy. Though the creep life is reduced when testing in the horizontal build direction, creep ductility was found to improve, which overcomes a major issue of most conventional ODS alloys. Extensive high-resolution characterization confirmed the presence of

type-A trigonal $Y_2O_3$ oxides in GRX-810. This contrasts with the more expected type-C cubic oxides present in the other NiCoCr-based ODS alloys explored in this study. The presence of trigonal oxides in GRX-810 is notable as it is the first observation of trigonal $Y_2O_3$ stable at ambient pressure and temperatures. This crystal structure appears correlated to a finer oxide dispersion present in GRX-810, resulting in a greater oxide number density and reduced inter-particle distance compared to ODS alloys that possess cubic $Y_2O_3$ particles. Ultimately, this unique oxide formation in GRX-810 results in significantly improved high-temperature properties. The insights and results presented here will hopefully promote the introduction of GRX-810 into extreme environment applications and provide a pathway to advance future ODS alloy development.

## Methods
### Materials
Pre-alloyed gas atomized GRX-810 powder feedstock with the composition published in ref. 31 was obtained from Linde Inc. The powders were sieved using -325 mesh (<53 μm). Nanoscale $Y_2O_3$ powder acquired from American Elements with <100 nm particle size was utilized for this study. $Y_2O_3$ particles were subsequently coated onto the GRX-810 base powder using a LabRAM2 high-energy acoustic mixer. Post-mixed powder was then sieved using a 230-mesh screen to remove any large oxide or metallic powder particles. GRX-810 samples were additively manufactured using a smaller laser powder bed fusion (L-PBF) EOS M100 machine (40 μm beam diameter) located at the NASA Glenn Research Center and a larger EOS M280 L-PBF machine (80 μm beam diameter) located at NAMPROS (Ardmore, AL). For GRX-810 builds on the EOS M280, a laser powder of 270 W, a laser speed of 1000 mm/s and a laser hatch spacing of 75 μm were used. All samples were then removed from the build plates using electrical discharge machining (EDM). Supplementary Fig. 12 provides an optical micrograph cross-section of as-built GRX-810, revealing a density greater than 99.9%. Supplementary Table 2 provides a list of all the alloys referenced in this study and their corresponding composition in weight percent.

### Characterization
For density measurements, cross-sections of 1 cm x 1 cm x 1 cm of as-built GRX-810 samples were sectioned parallel to the build direction

and polished using the procedure published in ref. 69 with a longer polishing time (20 min) during the 1 μm NAP step to better highlight smaller defects such as micro-cracks. Defect area fractions were determined using the software ImageJ[70]. For SEM analysis, samples were polished using SiC grit paper followed by 0.5 μm diamond suspension. High-resolution SEM imaging of the $Y_2O_3$ nanoparticles was performed using a Tescan MAIA3 in the ultra-high resolution (UHR) configuration at 15 kV. Three millimeter diameter STEM disc samples were extracted from metallographic samples of GRX-810 and ODS-ReB. The STEM samples were thinned down to 130 μm by hand using 600 grit SiC polishing paper. To achieve electron transparency, the polished STEM discs were electro-polished using a solution of 90% methanol and 10% perchloric acid at −40 °C and 12 V using a Struers twin-jet polisher. Microstructural analysis was performed on an FEI Talos at 200 kV using a HAADF detector. Atomic-scale characterization of the oxide dispersoids was performed using S-CORR probe aberration corrected and monochromated Thermo Fisher Scientific (TFS) Themis-Z STEM at an acceleration voltage of 200 kV. Atomic-scale imaging was performed on oxides located close to the hole to minimize an overlap with the surrounding fcc matrix. The acquired HRSTEM images and their corresponding FFTs were analyzed using SingleCrystal and CrystalMaker to identify the oxide structure. At least ten different oxides from each alloy were investigated to determine their crystal structure(s). High-resolution EDS data were collected by a Super-X energy dispersive X-ray spectroscopy detector in Themis-Z. The data were collected and processed using the TFS Velox software. In particular, the raw data in the original spectral maps were quantified using a standard Cliff-Lorimer (K-factor) fit (default k-factors available in Velox were used, as well as the Brown-Powell empirical ionization cross-section model), including background subtraction. The STEM micrographs were corrected for possible sample drift and scanning beam distortions using the drift-corrected frame integration function of Velox. Oxide size distributions were assessed through EDS maps using the Super-X detector of the Themis Z. A minimum of three distinct locations were analyzed for each alloy. The oxides were manually binarized by combining HAADF images with Yttrium element maps to accurately delineate oxides and determine their size distribution. Ex situ synchrotron-based HEXRD measurements were performed on GRX-810 and NiCoCr-ReB in the HIPed state at the experimental hutch (EH1) of the High Energy Materials Science beamline P07. The beamline is operated by Helmholtz–Zentrum Hereon at the Petra III synchrotron facility, German Electron Synchrotron[71]. For these HEXRD experiments, a beam energy of 87.1 keV (corresponding to a wavelength of 0.142 Å) was used, with less than 1% intensity contribution from radiation at half the wavelength. For analysis, the recorded Debye-Scherrer rings (Perkin Elmer XRD 1622 flat panel detector) were azimuthally integrated using pydidas[72] to generate line profiles for subsequent phase analysis. Differential scanning calorimetry was performed on FG GRX-810 using a Netzsch STA 409 in Ar.

## Mechanical testing

After printing, test coupons were removed from their respective build plates, select specimens completed HIP cycle at 1185 °C and 152 MPa. The HIP cycle also had the benefit of relieving residual stresses[73]. Both as-built and HIP specimens of GRX-810 were tensile tested at room and elevated temperatures using cylindrical specimens with a 3.175 mm diameter gage section. For the tensile tests at 1093 °C and below, the gauge length employed was 19.05 mm. For the elevated tensile tests above 1093 °C, the gauge length was 30.4 mm. The mechanical testing at 1093 °C and below was primarily performed at Metcut Research Inc., with some supplementary tests conducted at the NASA Glenn Research Center using samples that employed a gauge length of 19.05 mm. In both cases, tensile tests were performed at room temperature at a displacement rate of 0.127 mm/min for the first 1.5% strain and then increased to 1.016 mm/min until failure in general accordance

with the ASTM E8/E8M-21 standard. At 1093 °C and above, all tests were performed with a 0.305 mm/min strain rate until 4% strain was reached. At 4% elongation, the test was transitioned to displacement control at a rate equal to 1.98 mm/min ASTM E21-17 standard. Mechanical testing at 1093 °C and above was performed at Advanced Materials Testing & Technologies. These in-air high-temperature tests were performed using a 22-kip MTS servo-hydraulic load frame. Closed-loop test control was handled by MTS Multi-Purpose Test-Ware. Load was monitored by use of an MTS load cell having a capacity of 22 kip and resolution between 2 and 5% max load. Temperature was monitored using a Lumasense IMPAC ISR 12-LO two-color pyrometer with a range between 600 °C–3300 °C and a resolution of 1 °C. The pyrometer signal was used in a closed-loop temperature control housed within an Ambrell Easy Heat induction heating system. The induction heating system maintains the temperature of the specimen gage length to be within ±5 °C using a specially calibrated induction coil. The elevated temperature tests were performed in strain control up to 4% strain at a rate of 0.012 mm/mm/min according to ASTM E8. At 4% strain, the test was transitioned to displacement control until the specimen failed to give an approximate strain rate of 0.078 mm/mm/min. Creep tests were performed at 1093 °C by Metcut or at NASA Glenn Research Center in accordance with the ASTM E139-11 standard. Testing of creep samples was continued until rupture (unless otherwise stated), after which they were then rapidly air cooled to maintain the fracture surface. All specimens were tested along the build direction unless otherwise specified in the description. Some horizontal creep tests were performed using a high-throughput test setup utilizing a square cross-section, minimal machining, heating under load, and no extensometry. Such data has been in-family with traditional creep rupture lives but does not necessarily follow ASTM E193-11 and is plotted using a diamond symbol.

## Oxidation testing

Samples of AM718, 625, SC-180, B1900, B1900+Hf, ME3, CMSX-4, CMSX-10, and the FG GRX-810 alloy were prepared at nominally $25 \times 12.5 \times 3.5$ mm, which provided a total surface area of ~950 mm². Surfaces were polished to a smooth finish with 1 μm diamond paste. Cyclic oxidation was performed in a laboratory air tube furnace at 1093 °C in 1-h cycles at 1100 °C, 1200 °C, and 1300 °C for 200 h or until the sample failed. Cooling cycles between 1-h cycles occurred hanging in the lab air for 20 min. Samples were inserted into the hot zone at a rate of ~25 mm/s, reached 90% of the target temperature within 90 s, and were comfortably at the target temperature within 5 min. For the cooling cycle, the samples were removed at a rate of ~25 mm/s, and were below 500 °C within ~60 s and ~50 °C within 10 min. Sample weights were measured after several intervals for a total of 14 data points for each sample over the entire thermal exposure. For the 1300 °C cyclic oxidation tests performed on CMSX-4, CMSX-10, and GRX-810, the cycles performed for this test were carried out in a high-temperature air box furnace. Samples were inserted and removed from the furnace at a temperature to minimize the heating/cooling. After removal from the furnace between the cycles, samples were cooled on the bench in ambient air.

## Modeling

To investigate the relative stabilities of cubic, monoclinic, and trigonal $Y_2O_3$ phases, a series of DFT calculations with the VASP software package was performed[74–76]. PBE density functional[77] and the projector augmented wave method (PAW)[77,78] for all calculations were employed. The valence of Y and O atoms included the $4s^2 4p^6 4d^1 5s^2$ and $2s^2 2p^4$ electrons, respectively. Structural relaxations of $Y_2O_3$ used a plane-wave energy cutoff of 500 eV and gamma-centered k-point meshes of $2 \times 2 \times 2$ for the cubic phase and $1 \times 4 \times 2$ for the monoclinic phase. Testing showed that these settings achieved a convergence of <1 meV/atom. The criterion for stopping structural relaxations was to obtain

atomic forces less than 0.01 eV/Å. Supplementary Table 3 shows the results from the bulk $Y_2O_3$ structural relaxations. The calculated lattice parameters are overestimated but generally in good agreement with the experimental values. The calculated energy difference of 48.8 meV/atom confirms that the cubic phase is the ground state at 0 K. Changes to the chemistry of $Y_2O_3$ can affect the relative stability of its polymorphs. For example, it is well known that doping $Y_2O_3$ with Eu can favor the formation of the monoclinic phase[79,80]. Therefore, it is reasonable to hypothesize that alloying additions to GRX-810 could have a similar effect. To test this, we performed structural relaxations of Al, Ti, and Nb substitutional defects on the cation sites in both cubic and monoclinic $Y_2O_3$ and compared their relative energies. To allow for direct comparisons, the defects were inserted into 240 atom supercells (3 x 1 x 1 for cubic and 1 x 4 x 2 for monoclinic). We adjusted the k-point meshes accordingly to 1 x 2 x 2 for the cubic supercell and 2 x 2 x 2 for the monoclinic supercell. The valences of Al, Ti, and Nb atoms were $3s^23p^1$, $3d^24s^2$, $4p^64d^45s^1$, respectively. An important consideration for these calculations is the different cationic sites in $Y_2O_3$; the cubic phase has two distinct sites, and the monoclinic phase has three (see Supplementary Fig. 7)[81]. Therefore, structural relaxations were performed for substitutional defects at all five of these sites. We found that the two cationic sites in cubic $Y_2O_3$ were equivalent in terms of energy, but the monoclinic sites exhibited noticeable differences. All three substitutional species preferred to occupy the Y1 site in the monoclinic structure, with this also reducing the energy difference between the polymorphs. The change was most prominent for Al, which reduced the energy difference between polymorphs by 5 meV/atom, about 10% less than for bulk $Y_2O_3$. The site preference of the substitutional species can be explained by the different bonding environments at each site. Supplementary Fig. 12 shows the average Y-O bond length for each cation site in the relaxed cubic and monoclinic structures. For the cubic phase, there is no noticeable difference in bonding between the sites. For the monoclinic phase, however, the Y1 sites have significantly shorter bonds than the Y2 and Y3 sites (4.6% shorter on average). Considering the substitutional cations $Al^{3+}$ (67.5 pm), $Ti^{3+}$ (81 pm), and $Nb^{3+}$ (86 pm) all have smaller radii than $Y^{3+}$ (104 pm)[82], this provides a plausible explanation as to why these defects prefer the Y1 site and make the monoclinic phase more energetically favorable. We used these insights into site preference to add new solute atoms. After relaxation of a defect-containing supercell, the next substitutional atom was added at the cation site with the shortest average Y-O bonds. In this way, the substitutional defects were placed at likely energetically favorable sites without needing to employ a computationally expensive search. This also provides validation to our approach for generating the defected supercells, as each additional substitutional atom resulted in a comparable reduction of the energy difference, demonstrating that it was indeed placed at an energetically favorable site. AIMD simulations were performed on 1 x 1 x 1 (80 atom) and 4 x 4 x 3 (240 atom) supercells of cubic and trigonal yttria, respectively. The trigonal yttria supercell was transformed to an orthogonal one to aid in the data analysis. The same VASP settings described for 0 K calculations were employed, with 2 x 2 x 2 and 1 x 1 x 1 k-point meshes for the cubic and trigonal supercells, respectively. Additional AIMD settings included a Langevin thermostat and barostat with friction coefficients of 10 $ps^{-1}$ and a fictitious mass of 1000 amu, and an MD timestep of 2 fs. Initial lattice parameters were determined via short (5 ps) NPT AIMD simulations. Subsequently NVT simulations were carried out for 20 ps for stability and diffusion calculations. Stability was determined according to elastic constants, while diffusion was evaluated according to averaged mean squared displacements.

## Data availability

The experimental data that support the findings of this study are available from the corresponding author on request. Source data are provided with this paper.

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

## Acknowledgements

Funding for this study was provided by NASA's Aeronautics Research Mission Directorate (ARMD)—Transformational Tools and Technologies (TTT) Project Office and NASA's Space Technology Mission Directorate (STMD) Game Changing Development (GCD) Program under the optimized and Repeatable Components in Additive Manufacturing (ORCA) project. J.M., A.B., and M.J.M. acknowledge the support of the National Science Foundation and the DMREF program under grant #2323717. The authors also acknowledge DESY (Hamburg, Germany), a member of the Helmholtz Association HGF, for the provision of experimental facilities. Parts of this research were carried out at PETRA III. T.M.S. would like to acknowledge Jamesa Stokes for performing the DSC analysis on GRX-810. For more information on this technology, and to discuss licensing and partnering opportunities, please contact grc-tech-transfer@mail.nasa.gov and reference LEW-19886-1 and LEW-20020-1.

## Author contributions

T.M.S. wrote the manuscript. T.M.S., C.A.K., T.P.G., and P.R.G. designed the experiments and performed the microstructural/mechanical characterization. T.M.S., A.B., J.M., M.H., and M.J.M. performed the TEM analysis. T.M.S. and A.J.W. produced the powder feedstock—coated and uncoated. A.C.T. and A.J.W. operated the EOS M100 and developed the build parameters. B.J.H. and B.J.P. performed the cyclic oxidation tests. G.P., M.I.M., and J.W.L. performed the DFT models. A.B., A.S., and S.N. performed the synchrotron experiments.

## Competing interests

The authors declare no competing interests.
