## [Transparent Peer Review file · Nature Communications]

The mechanisms underlying the enhanced high-temperature properties of GRX-810

Corresponding Author: Dr Timothy Smith

Version 0:

Reviewer comments:

Reviewer #1

(Remarks to the Author)

Review of The ultra-high temperature stability and properties of GRX-810

The authors provide a manuscript with an intriguing and important result – 50 MPa of strength in an alloy at 98% of its melting point. This result is attributed to the presence of fine-scale yttria nano-dispersoids. The reviewer is completely unconvinced by the result, which is not supported with sufficient experimental evidence or analysis. Specific issues that would make this article publishable that need to be addressed are listed below.

1. First, no information on the composition of the alloy powder is given, this is essential in the main body of the manuscript.
2. What is meant by 98% of melting temperature? Is it 98% of the solidus? The liquidus?
3. It is odd that the main result, 50 MPa at 1300°C is not accompanied by a stress strain curve at that temperature in Figure 2. The shape of the stress strain curve even at 1093°C does not convey useful strength, as softening occurs immediately upon loading. What is the source of the softening? Also, the plots do not contain any hatch marks.
4. Testing at 1300°C is extremely challenging, however, in the methods section there is no information that conveys how these challenges were surmounted. Was the testing in vacuum or air? What extensometry was used? What materials were used for gripping? What were the sample dimensions and were they heated end to end or only in the gage section? How much did the temperature vary along the gage length? Were all the details of the ASTM standard followed during testing?
5. In Figure 3, MPa is not properly capitalized in all of the plots.
6. If the oxides are the source of strengthening above 1200°C, a simple Orowan calculation assuming a shear modulus of 10GPa would suggest an oxide spacing of approximately 50nm. Looking at the micrograph in Fig. 6, the diameter of the oxide is about 50nm. This would suggest an extraordinarily high volume fraction of oxide is needed to achieve this strength. The potential mechanisms of strengthening must be addressed in the context of the high temperature strength. This should be supported with information on the size, distribution and volume fraction of the oxides.
7. Line 134 – why is it necessary to “presume” that CMSX-4 recrystallizes when one could simply section the sample and verify after testing? What did the structure of the GRX-810 samples look like after testing, especially at the highest temperatures? Such investigations would seem to be essential to validating the testing methodology and results.
8. The authors should point out that the oxidation weight changes are on a g/cm² scale rather than mg/cm² scale, which would be typical. They refer here to B1900 as a single crystal alloy; while it has been grown as a single crystal it is certainly not used in that capacity due to its poor high temperature properties. Why have the authors chosen at this point to abandon the comparison with CMSX-4 or any other commercial single crystal alloy? The oxidation data show uniform weight loss over the entire range of conditions investigated. The authors neglect to mention that chromia is volatile at these temperatures and so this behavior would be expected. There is a good possibility that any oxides observed on the surface formed during cooling of the samples from elevated temperatures; this must be addressed.
9. In Figure 6 the authors again conveniently fail to compare to single crystal alloys (CMSX-4/10, René N5/N6 or PWA 1480/1484. This is a lot of published literature on these alloys at 1093°C and at even higher temperatures that demonstrates these alloys are more creep resistant at this temperature. Some comparisons on proper single crystal alloys must be included.
10. Fig. 5 – it appears that the GRX-810 has recrystallized after 10 one hour cycles at 1300°C, with a large increase in grain size. This is not addressed anywhere in the paper in the context of 1300°C properties. Also, the small differences in the initial grain size are not likely to be responsible for differences in the properties – are there slight differences in chemistry between the fine and coarse grain material?

11. While the reviewer realizes there is limited space, the microscopy does not seem to conclusively show the transformation in the oxide crystal structure. While OK calculations are useful, the authors have made no attempt to look into the extensive literature on high temperature phase equilibria in oxides to justify the monoclinic structure. This is a major shortcoming.
12. There is insufficient detail on the acoustic mixing of the powder. Time of mixing and details of acoustic settings?
13. The authors also do not sufficiently address the issue of melting the oxides during printing. It would be helpful to give the melting temperature of yttria and refer to any literature on DTA characterization of ODS alloys.

Reviewer #2

(Remarks to the Author)

In this manuscript, the authors build upon their previous work (Nature, 2023, 617(7961): 513-518.) by further evaluating and investigating the high-temperature performance of 3D-printed GRX-810 alloy at elevated temperatures (1100 to 1300°C), including its creep properties and oxidation behavior. The material demonstrates excellent high-temperature performance, particularly at temperatures as high as 1300°C, surpassing some existing high-temperature materials. However, this paper appears to be more of a data supplement to the previous study rather than presenting significant innovation in terms of material design, preparation, or research methodology. Additionally, the authors have not provided clear cause-consequence relationship between the oxides and the alloy properties. While they suggest that the introduction of Nb affects the oxide structure of Y, this explanation does not seem sufficient to account for the enhanced performance of the alloy. Therefore, major revisions should be needed for improving this paper. My detailed comments are as follows:

1. The authors need to provide further clarification regarding the novelty of this paper, particularly in relation to their previous work. In addition to expanding the temperature range for testing, they should clearly differentiate this study from the earlier one and explain the new insights or impacts it brings. Regarding the discussion on Y-based oxides, based on the authors' statements, the section primarily reflects how the alloy system affects the oxides, rather than how the oxides contribute to the overall performance of the alloy. This distinction needs to be addressed to better demonstrate the significance of the oxide's role in enhancing alloy performance.
2. The authors have compared the alloy with several other alloys; however, to demonstrate the effectiveness of the oxides, the most direct comparison would be with the base material of the same composition without oxides (as was done in the authors' previous work). Unfortunately, this critical comparison is missing in the current manuscript.
3. Page 12: "After 10 cycles, GRX-810 has an oxide layer on the surface of the sample that is 10-20 μm thick, while the 718 sample was oxidized completely through the bulk of the material. 718 also exhibited large porosity and grain growth as shown in 6(a)." It should refer to Figure 5(a).
4. Page 14: "It is possible that the presence of nano yttria particles in the scale may also assist in adhesion or some other mechanism that assists in oxidation performance similarly to how the inclusion of Y_2O_3 in the base alloy provides the benefit of inhibiting grain growth in the metal matrix⁴⁵." It is necessary to further clarify or modify the statement regarding why "assist in adhesion" is similar to the grain growth inhibition mechanism seen in ODS (oxide dispersion strengthened) metals. Typically, grain growth inhibition is associated with mechanisms related to Zener pinning.
5. Page 19: "Figure 7 shows horizontal creep data for two types of GRX-810 specimens: cylindrical specimens printed on the EOS M280 with 25x recycled powder, and flat specimens printed with freshly mixed powder on an EOS M100." The 25x recycled powder still retains certain properties, it is good. However, for a rigorous comparison, the horizontal creep data from the same powder should be provided.

Reviewer #3

(Remarks to the Author)

This manuscript presents a comprehensive study on the ultra-high temperature stability and mechanical properties of the NASA-developed alloy GRX-810, which is specifically designed for AM applications in extreme environments. The paper outlines a series of high-T tensile, creep, and cyclic oxidation tests conducted at temperatures up to 1316°C, providing valuable insights into the alloy's performance. The authors explore how GRX-810 outperforms conventional superalloys and highlight the benefits of the oxide dispersion strengthening (ODS) mechanism that has been incorporated through a novel powder coating process.

The manuscript is an extension of previous work by the authors (Smith, T. M. et al., Nature 617, 513–518, 2023), which introduced GRX-810. While this paper presents additional data on microstructure, nano oxide formation, and high-temperature performance, it fails to sufficiently clarify the reasons for the alloy's superior performance at high temperatures compared to other alloys. As such, I do not believe the results in this manuscript are impactful enough to justify publication in Nature Communications. A more detailed feedback can be found below:

General remarks:

- 1) Including page numbers would have made reviewing the manuscript easier
- 2) While the current manuscript presents more data, it seems that many results shown in Figures 3-5 have already been

published. These should be clearly indicated as such. In my opinion, the novelty of the results in this manuscript is not significant enough to warrant a separate publication.

3) The quality of several figures needs improvement. In particular:

- a) The font size in Figures 1, 2, and 3 is too small and should be enlarged for readability.
- b) The axis labels in Figure 4 are hard to read and should be more prominent.
- c) The choice of yellow and light orange colors in some graphs is in my opinion, not ideal, making it difficult to distinguish data points.
- d) In Figure 3, the term "Mpa" should be corrected to "MPa."

Specific comments:

4) It is unclear in what condition (e.g., fine grain (FG) or coarse grain (CG), as-built or HIPped) the samples were tested for cyclic oxidation. Additionally, while the authors mention the excellent oxidation performance of GRX-810, the discussion on this topic is quite superficial. Currently, the focus on cyclic oxidation is overshadowed by the detailed discussion of creep behavior and nano oxide evolution, leaving key insights about the material's long-term oxidation stability unexplored. Expanding on this topic with more in-depth discussion would add considerable value to the manuscript.

5) The authors should provide a more thorough evaluation of their creep data. Specifically:

- a) Present the minimum creep rates as a function of applied stress for both vertical and horizontal orientations and derive the stress exponent n by plotting the $\log(\text{minimum strain rate})$ vs. $\log(\text{stress})$ and applying Norton's creep law. This analysis would provide insights into the dominant creep mechanism.
- b) A microstructural analysis of the creep samples near the fracture surface would also help in understanding the failure mechanisms, especially in relation to the role of nano-dispersoids as well as IM precipitates or elemental segregation on grain boundaries.

6) The authors mention the well-known phenomenon of creep anisotropy in conventionally manufactured ODS alloys. However, it should be noted that compressive creep anisotropy has also been recently observed in ODS Hastelloy X and ODS Ni-Cr-Al-Ti model alloys, both fabricated via LPBF (<https://doi.org/10.1016/j.addlet.2022.100069>, <https://doi.org/10.1002/adem.202200753>). For the Hastelloy X, the compressive creep stress applied perpendicular to the aligned, textured grains resulted in significantly faster creep in both ODS and non-ODS variants. This suggests that the interplay between texture, grain shape, grain size, and intermetallic grain boundary precipitates is complex, which is likely true for GRX-810 as well. The authors are recommended to incorporate these findings and/or extend the microstructural characterization (see points 7) and 8)) to provide a more nuanced discussion of GRX-810's anisotropic behavior.

7) The manuscript would benefit from a more in-depth study of the nano oxides and their impact on the high-temperature performance of GRX-810. As it stands, the discussion does not offer substantial new knowledge beyond the previous publication - even the authors acknowledge that various important questions remain still unanswered. The authors' sub-chapter on nano-oxide characterization focuses on the formation of monoclinic Y_2O_3 , probably stabilized by Nb, but does not convincingly explain how this affects the alloy's high-temperature strength or creep performance. The discussion remains speculative, and the manuscript would benefit from stronger evidence linking these observations to the alloy's mechanical properties. More emphasis on how the nano-oxides contribute to strength, oxidation resistance, and creep performance is needed to justify the paper's novelty.

8) In the same chapter, the authors write: "The finding that oxides transform from their initial structure, and their interaction with other elements from the metal matrix, strongly indicate that the oxides melt and reprecipitate during the AM process, which has been hypothesized in recent published work". This statement has already been demonstrated in previous work by other researchers, such as Kenel et al., who observed a similar phenomenon during laser powder bed fusion of an ODS Ni-Cr-Al-Ti γ/γ' model alloy. Their study showed the decomposition of Y_2O_3 at the high temperatures during LPBF, leading to Y segregation to grain boundaries and the formation of Y-Ni IM precipitates within grains and on grain boundaries (besides formation $Y_4Al_2O_9$ slag because of the high amount of Al in their alloy (<https://doi.org/10.1016/j.addma.2021.102224>). Given the composition of GRX-810, similar behavior is expected during LPBF processing, and this might affect the creep performance. Thus, I encourage the authors to further investigate the fate of the Y_2O_3 dispersoids during LPBF, the grain boundary properties and their potential influence on creep behavior.

Version 1:

Reviewer comments:

Reviewer #1

(Remarks to the Author)

The authors have been partially responsive to the reviewers, even adding some additional experiments. However, there are still a number of issues that must be addressed before this paper is publishable

1. Fig. 1: The authors still have not addressed the softening in the 1093C tensile tests. Were the samples sectioned after testing? Did the creep samples have similar instabilities? Was there recrystallization during tensile testing but not creep testing?
2. Fig. 1: What is the explanation for the 1260C and 1316C samples having essentially the same flow stress in spite of a

temperature difference of 56C? What are the serrations in the stress strain curves?

3. One should not have to go to supplementary materials to find the composition of this material. Where the chemical measurements on the powder or the printed material? Are there any differences in oxygen in the fine vs coarse grained material?
4. Fig. 3. How reproducible are the creep results: is the fine grain vs coarse grain effect consistent, given that it is such a small difference in grain size?
5. If the solidus is only predicted, any claims about strength relative to the solidus should be removed. Is there any DTA data?
6. The authors still have not added any convincing discussion about the mechanisms of strengthening up to 1316C. The ODS / newly identified carbide strengthening do not seem consistent with the creep stress exponents. Supplementary Fig. 10 (which should have testing conditions listed) should be helpful for this discussion (comparisons of the radii of dislocation loops with applied stress, particle spacing etc) which should appear in the main body.
7. On page 11, the authors claim that the oxidation performance must be good since the sample survived tensile testing; however at a rate of 2×10^{-4} /s for 4% strain the test would last approximately 3 min, so this argument should be removed.
8. The authors show no evidence of a Cr₂O₃ film forming at 1300C even in the short term, with only a very low magnification image in Fig. 6. The linear weight loss is a classic sign of vaporization. Unless chromia scales are actually shown and analyzed, these arguments about protective chromia are completely unsupported. The weight loss, given in g/cm², is quite high.
9. In Fig. 8, the Larson Miller parameter has the wrong units (time units are missing).
10. Testing has been conducted on fine grain, coarse grain, horizontal and vertical with an additional HIP cycle. Every sample and figure should indicate which of these material parameters are being addressed.
11. The mechanical testing at 1093C and above is not clearly described in this section – was it Metcut or AMTT? The gage lengths of the specimens have still not been given.
12. Supplementary Table 1 – is all data for the same strain rate?

Reviewer #2

(Remarks to the Author)

The reviewer continues to find the discussion on yttrium oxide particles, particularly the potential mechanisms by which the novel crystal structure emphasized by the authors enhances high-temperature performance, insufficiently explored. In the revised manuscript, the authors included a simple statistical analysis of oxide sizes, suggesting that the trigonal structure may result in finer particle sizes compared to conventional oxides. However, this section is inadequately developed and requires expansion. For instance, what is the specific impact of these particles on high-temperature creep behavior? Given that creep in this material is primarily governed by grain boundary sliding, how does the oxide affect this mechanism? Is the substantial performance enhancement solely due to smaller particle sizes? Are there differences in the spatial distribution of yttrium oxide particles compared to other ODS alloys? Additionally, are other factors, such as interfacial effects, contributing to the observed improvements?

Regarding the differentiation of this work from their previous study (Nature, 2023, 617(7961): 513-518.), the authors highlight their discovery and structural characterization of the novel yttrium oxide crystal structure, with additional distinctions limited to higher testing temperatures, different stress conditions, and varying orientations for alloys. The reviewer considers this level of novelty insufficient. Moreover, if the oxide crystal structure is deemed critical to material performance, why is this aspect not mentioned in the abstract or introduction? The authors are strongly encouraged to explicitly emphasize the significance of the new crystal structure in these sections to enhance the manuscript's coherence and highlight its innovation.

Reviewer #3

(Remarks to the Author)

The main reviewer questions have been addressed and the manuscript has been revised accordingly. It may now be accepted for publication in Nature Communications

Version 2:

Reviewer comments:

Reviewer #1

(Remarks to the Author)

This manuscript has been substantially improved, with the additional material on oxide characterization, strengthening calculations and other details of materials and testing added. Nearly all reviewer concerns have been addressed. The only area not adequately addressed relates to oxidation.

In the text, the authors refer to “one hour isothermal cycling”. Since it refers to a cyclic oxidation experiment, this should be re-worded to report heating and cooling rates and the fact that one hour at temperature was employed. The GRX 810 loses weight (on the g/cm³ scale) at all temperatures – 1100, 1200 and 1300°C. The authors state that: “Superalloys 625 and GRX-810 completed the full 200 1-hr cycles and remained intact without extensive weight loss or distortion.” The reviewer suggests that the “without extensive weight loss” is not justified. Comparing to B1900 at 1200°C, is not really helpful since

this alloy was never designed to operate near that temperature. The presence of a chromia layer following cooling is used to conclude "oxide stability near melting". This layer was likely strongly influenced by the cooling cycle. If the authors want to claim that chromia is thermodynamically stable at 1300°C, then they should justify this with thermodynamic calculations.

Reviewer #2

(Remarks to the Author)

The authors have addressed the comments well in the revised Paper. Therefore, this paper can be accepted.

Version 3:

Reviewer comments:

Reviewer #1

(Remarks to the Author)

This manuscript is now ready for publication.

Dr. Timothy Smith
NASA Glenn Research Center
21000 Brookpark Rd.
Cleveland Oh, 44145
216-433-2632
timothy.m.smith@nasa.gov

Dear Editors and Reviewers,

We are very excited to have been given the opportunity to revise our manuscript, "The ultra-high temperature stability and properties of GRX-810" for Nature Communications. We have carefully considered the comments and concerns made by all three reviewers and in the list below describe how each was addressed. We also thank everyone for the patience as we worked to address these concerns. We have added significantly more analysis and characterization in the revised version of this paper and feel it is a much stronger study as a result. We also hope that the additional time, resources, and characterization we invested in this revision highlights how seriously we considered each of the reviewer's concerns. We are also greatly appreciative of the effort and time it takes the reviewers to provide their helpful insights. Our responses to each of their comments are provided below.

Reviewer 1:

- 1. First, no information on the composition of the alloy powder is given, this is essential in the main body of the manuscript.**
 - The composition of GRX-810 for both FG and CG versions, along with all the other alloys tested, are given supplementary table 2 and we feel that this is adequate for the paper. However, if reviewer 1 and the editors feel the composition of GRX-810 needs to be provided in the body of the manuscript we will do so.
- 2. What is meant by 98% of melting temperature? Is it 98% of the solidus? The liquidus?**
 - The sentence was adjusted to clarify that this is based on the alloys predicted solidus temperature.
- 3. It is odd that the main result, 50 MPa at 1300°C is not accompanied by a stress strain curve at that temperature in Figure 2. The shape of the stress strain curve even at 1093°C does not convey useful strength, as softening occurs immediately upon loading. What is the source of the softening? Also, the plots do not contain any hatch marks.**
 - We now include stress strain curves for both vertical and horizontal tensile at 1316°C and 1260°C respectively, as shown in the figure below. Though the alloy does show softening at 1093°C, referring to the strength as not useful seems strange considering the paper also shows phenomenal creep lives at this temperature regime. Clearly, the alloy still possesses useful strength at this temperature based on the creep response. Many, if not most, alloys exhibit strain softening at these temperatures due to dynamic recrystallization and recovery processes, so this response is not unusual.

Fig. 1: Mechanical tests of vertical and horizontal GRX-810. (a) room temperature, (b) 1093°C, and (c) Extreme temperature (1260/1316°C) tensile curves of HIP'd GRX-810 in the vertical and horizontal directions.

4. **Testing at 1300°C is extremely challenging, however, in the methods section there is no information that conveys how these challenges were surmounted. Was the testing in vacuum or air? What extensometry was used? What materials were used for gripping? What were the sample dimensions and were they heated end to end or only in the gage section? How much did the temperature vary along the gage length? Were all the details of the ASTM standard followed during testing?**

- The following description of the elevated tensile tests were added and are shown below.

“In air mechanical testing at 1093°C and above was performed at Advanced Materials Testing & Technologies (AMTT). These high temperature tests were performed using a 22 kop MTS servo hydraulic load frame. Closed-loop test control was handled by MTS multi-purpose test ware. Load was monitored by use of an MTS load cell having a capacity of 22 kip and resolution between 2% and 5% max load. Temperature was monitored using a Lumasense IMPAC ISR 12-LO two-color pyrometer with a range between 600C-3300C and a resolution of 1C. The pyrometer signal was used in a closed loop temperature control housed within an Ambrel Easy Heat induction heating system. The induction heating system maintains temperature of the specimen gage length to within +/- 5C using a specially calibrated induction coil. The elevated temperature tests were performed in strain control up to 4% strain at a rate of 0.012 mm/mm/min according to ASTM E8. At 4% strain the test was transitioned to displacement control until the specimen failed to give an approximate strain rate of 0.078 mm/mm/min.”

5. **In Figure 3, MPa is not properly capitalized in all of the plots.**

- This has been fixed for all updated figures in the revised version. Thank you for bringing that to our attention.

6. **If the oxides are the source of strengthening above 1200°C, a simple Orowan calculation assuming a shear modulus of 10GPa would suggest an oxide spacing of approximately 50nm. Looking at the micrograph in Fig. 6, the diameter of the oxide is about 50nm. This would suggest an extraordinarily high-volume fraction of oxide is needed to achieve this strength. The potential mechanisms of strengthening must be addressed in the context of the high temperature strength. This should be supported with information on the size, distribution and volume fraction of the oxides.**

- Significant effort and resources were included in the revision to better characterize the oxides found in GRX-810 as shown in the updated figure 7 below.

Fig. 7: High-resolution characterization of oxide nano-particles. Atomic-scale STEM micrographs of oxides found in GRX-810 (a) revealing faceted oxide faces and (b) trigonal crystal structure. (c) The fast fourier transformation from (b) supporting the trigonal crystal structure. (d) A comparison of the oxides in NiCoCr-ReB and GRX-810 using synchrotron-based high-energy X-Ray diffraction confirming the presence of trigonal oxides in GRX-810 and cubic oxides in NiCoCr-ReB ODS. (e) DFT calculations for the energy difference between cubic, monoclinic, and trigonal phases of Y_2O_3 as a function of substitutional defect concentration. (f) The size distribution of oxides in NiCoCr-ReB ODS and GRX-810 using high resolution scanning transmission electron microscopy.

We were able to show that the oxides in GRX-810 are indeed significantly finer compared to the other NiCoCr-based ODS alloys tested in this study. This finer dispersion is discussed with respect to the different oxide crystal structure found stable in GRX-810 compared to the more expected cubic structure found in NiCoCr-ODS. Clearly, improvements in strength would be expected by this finer dispersion. However, the strength alone cannot be calculated by the oxide characteristics alone. In the revision we also show that nano-scale carbides precipitate during creep and thus further add to the dispersion strengthening of GRX-810 (Supplementary Figure 10). In addition, grain boundaries also contribute significantly to properties at these high temperatures and must be considered. We hope that the additional characterization and analysis better explain the properties we reveal in the revised study. We would like to make note that quantifying the volume fraction of these oxides has been quite difficult so far. We have employed many different high-resolution techniques and the determined error associated with each technique made the calculated value unreliable. Even the high-energy synchrotron couldn't provide reliable volume fraction values.

7. Line 134 – why is it necessary to “presume” that CMSX-4 recrystallizes when one could simply section the sample and verify after testing? What did the structure of the GRX-810 samples look like after

testing, especially at the highest temperatures? Such investigations would seem to be essential to validating the testing methodology and results.

- The CMSX-4 tensile strength results are from cited work. We do not have the samples in hand to characterize. Still, we removed this statement as it was speculative and not supported by any evidence.
 - “This result is presumably a consequence of the strengthening precipitates in CMSX-4 dissolving at these temperatures which severely reduces the strength of the single crystal³³”
- 8. The authors should point out that the oxidation weight changes are on a g/cm² scale rather than mg/cm² scale, which would be typical. They refer here to B1900 as a single crystal alloy; while it has been grown as a single crystal it is certainly not used in that capacity due to its poor high temperature properties. Why have the authors chosen at this point to abandon the comparison with CMSX-4 or any other commercial single crystal alloy? The oxidation data show uniform weight loss over the entire range of conditions investigated. The authors neglect to mention that chromia is volatile at these temperatures and so this behavior would be expected. There is a good possibility that any oxides observed on the surface formed during cooling of the samples from elevated temperatures; this must be addressed.**
- Many new results and analysis have been included in the revised paper to respond to these concerns. See below to the added work that was included to respond to this concern:
 - “To better understand the oxidation results, additional 1300°C cyclic oxidation tests were performed comparing GRX-810 and the current state-of-the-art single crystal blade alloys CMSX-4 and CMSX-10 as shown in Figure 6(a).

Fig. 6: Oxidation and microstructural stability at 1300°C. (a) Additional oxidation results comparing GRX-810 to commonly used single crystal blade alloys at 1300°C. There is an additional GRX-810 test showing the weight change after a 24-hour soak at 1300°C. Metallography cross-sections of (b) superalloy 718 after 7 cycles and (c) GRX-810 after 10 cycles.

The results in Figure 6(a) appear to further confirm the oxidation performance found in Figure 5(c). GRX-810 had a slower weight loss rate compared to CMSX-10. CMSX-4 had the best oxidation performance over the 20 cycles measured; however, after 15 cycles its weight loss rate started to accelerate and exceed that of GRX-810. This change may be caused by a depletion of Al near the surface compromising the alloy's ability to maintain a stable alumina oxide layer. In addition to the cyclic tests, a 24-hour soak at 1300°C was performed on GRX-810 to elucidate the cause of weight loss in the GRX-810 samples at this temperature. The sample was found to gain significant weight during the soak suggesting that the weight loss found in the cyclic tests was dominated by oxide scale spallation upon cooling and not volatilization of chromia at elevated temperatures."

9. In Figure 6 the authors again conveniently fail to compare to single crystal alloys (CMSX-4/10, René N5/N6 or PWA 1480/1484. This is a lot of published literature on these alloys at 1093°C and at even higher temperatures that demonstrates these alloys are more creep resistant at this temperature. Some comparisons on proper single crystal alloys must be included.

- Figure 6 has been removed and replaced with better creep analysis, including Larson miller parameter and calculated stress exponent for GRX-810 in the horizontal and vertical orientations as shown below.

“Orientation effects on GRX-810 creep properties

The creep curves provided in Figures 3 and 4 reveal the influence that grain structure has on the creep performance in GRX-810. An overview of these results with additional creep tests are highlighted in Figure 8.

Fig. 8: Creep Strength Overview of GRX-810. (a) The Larson Miller Parameter vs. stress for creep rupture in GRX-810 in the horizontal and vertical build directions compared to plate Haynes 230. (b) Creep strain rate vs. stress at 1093°C for both orientations and the calculated stress exponents.

The Larson Miller Parameters revealed in Figure 8(a) was calculated using the equation below.

$$Rupture\ LMP = (Temp\ (C^{\circ}) + 273.15) * (\log t_{hr} + 20) \tag{1}$$

Clear differences in creep rupture life of GRX-810 was observed between the vertical and horizontal build orientations. When designing parts for GRX-810 this creep anisotropy should be considered for complex stress states, however preliminary component demonstrations have not shown unexpectedly poor performance for GRX-810³². Using the same creep tests, the creep stress exponents were calculated from the slope of the minimum strain rate as a function of stress for both orientations as shown in figure 8(b) using the below equation⁵⁷.

$$n = d(\ln(\dot{\epsilon}))/d(\ln\sigma) \tag{2}$$

The stress exponents reveal that the two orientations may be under different creep deformation failure modes. Higher values suggest dislocation creep is dominant while lower values suggest that diffusion creep and grain boundary sliding are more active. The results in Figure 8(b) suggest that grain boundary sliding mechanisms may be more prominent in horizontal oriented creep compared to the vertical orientation in GRX-810. This result is supported by the fracture surfaces characterized in Supplementary Figure 8. The fracture surface found in the horizontal GRX-810 creep specimen reveals that failure

occurs primarily along grain boundaries. In contrast, the vertical GRX-810 creep specimen did not clearly appear to fail along grain boundaries.

- With this analysis any mechanical engineer and/or designer will be able to quickly compare GRX-810 to both cast and wrought superalloys as well as single crystal blade alloys. However, I, the corresponding author, would like to make it clear to reviewer 1 that at no point has this group claimed GRX-810 is superior or even close to the creep strength of single crystal blade alloys. We make this clear in every public presentation or when asked. GRX-810 was produced to replace AM superalloys such as 625, 718, and H230 or the Nb-based alloy C103 at its lower temperature limits. This is why our studies always compare the results to these alloys. It's what our industry and collaborators are most interested in. We include the comparison to single crystals in the oxidation study because they are superalloys that primarily employ a protective alumina scale.

10. Fig. 5 – it appears that the GRX-810 has recrystallized after 10 one-hour cycles at 1300°C, with a large increase in grain size. This is not addressed anywhere in the paper in the context of 1300°C properties. Also, the small differences in the initial grain size are not likely to be responsible for differences in the properties – are there slight differences in chemistry between the fine and coarse grain material?

- New EBSD analysis and discussion have been included to highlight the recrystallization that occurs in GRX-810 at 1300C after 10 cycles.
“The grain orientation map in Supplementary Figure 2(c) reveals recrystallization of the grain structure after 10 thermal cycles at this temperature.”

Supplementary Figure 2: SEM micrographs of the oxide (fine dark circular features) size and morphology of (a) as-printed GRX-810 and (b) GRX-810 after ten 1300°C 1-hour cycles. Grain orientation map of GRX-810 after ten 1300°C 1-hour cycles revealing a recrystallized grain structure.

11. While the reviewer realizes there is limited space, the microscopy does not seem to conclusively show the transformation in the oxide crystal structure. While OK calculations are useful, the authors have made no attempt to look into the extensive literature on high temperature phase equilibria in oxides to justify the monoclinic structure. This is a major shortcoming.

- We thank the the reviewer suggesting we relook at the oxide characterization because it resulted a correction to our previous analysis. Honestly, responding to this concern probably contributed to the largest delay in resubmitting the paper. As the reviewer will find significant and state-of-the-art characterization was performed to better characterize the oxides. We confidently reveal that the oxides

are actually type-A trigonal Y_2O_3 instead of monoclinic. Multiple zone axis were characterized to confirm this as shown below.

- **Supplementary Figure 6: XRD analysis of coated GRX-810 powder.**

Lastly, we also performed high energy synchrotron to confirm this result.

12. There is insufficient detail on the acoustic mixing of the powder. Time of mixing and details of acoustic settings?

- We are forbidden by the US government to publish this data due to export restrictions.

13. The authors also do not sufficiently address the issue of melting the oxides during printing. It would be helpful to give the melting temperature of yttria and refer to any literature on DTA characterization of ODS alloys.

- We feel the finding that the oxides in GRX-810 have changed crystal structure during the printing process highly suggests the oxides have melted. Still, further analysis on this subject will be a focus in future publications and is out of scope of this study.

Reviewer 2:

- 1. The authors need to provide further clarification regarding the novelty of this paper, particularly in relation to their previous work. In addition to expanding the temperature range for testing, they should clearly differentiate this study from the earlier one and explain the new insights or impacts it brings. Regarding the discussion on Y-based oxides, based on the authors' statements, the section primarily reflects how the alloy system affects the oxides, rather than how the oxides contribute to the overall performance of the alloy. This distinction needs to be addressed to better demonstrate the significance of the oxide's role in enhancing alloy performance.**
- We hope reviewer 2 will find that the new version of the paper strongly separates itself from the previous study. Below is a list of major changes we have made to the revision to better highlight this fact.
 - o New tensile results highlighting the difference between build orientations up to 1316°C.
 - o Horizontal creep performance of GRX-810
 - o High resolution characterization of oxides in GRX-810 using STEM, STEM-EDS, synchrotron diffraction, DFT, and XRD. This analysis confirms the presence of trigonal Y_2O_3 oxides in GRX-810. This is the first observation of stable trigonal Y_2O_3 found in ambient temperature and pressures.
 - o Larson Miller parameter and calculated stress exponents for creep in GRX-810 for both vertical and horizontal build orientations.
 - o We also removed the original figure 6 that was an updated version from the previous paper.
- 2. The authors have compared the alloy with several other alloys; however, to demonstrate the effectiveness of the oxides, the most direct comparison would be with the base material of the same composition without oxides (as was done in the authors' previous work). Unfortunately, this critical comparison is missing in the current manuscript.**
- This analysis has now been included with respect to creep as shown below in Supplementary Figure 9 and corresponding analysis.

Supplementary Figure 9: As-built horizontal creep rupture lives vs stress for GRX-810 with and without oxides. The diamond marks the creep rupture life of as-built non-ODS GRX-810 in the vertical orientation.

“Another study was performed comparing the rapid rupture creep of GRX-810 with and without the addition of oxides in the horizontal loading direction. The results are shown in Supplementary Figure 9. The results reveal that, even in the inferior horizontal orientation, the oxides are providing large improvements in creep life (two orders of magnitude longer) when compared to GRX-810 printed using powder without the oxide loading. Interestingly, the GRX-810 material tested in Supplementary Figure 9 possessed a fine grain microstructure (due to being manufactured on the smaller M100 AM printer) and appears to perform better in creep at lower stresses than does the CG GRX-810 material shown in Figure 4. Thus, the weaker creep life of GRX-810 in the horizontal orientation is a direct result of the detrimental grain structure with relation to the stress axis and not due to the addition of the nano-oxides. This is further confirmed by the lower creep life possessed by as-built non-ODS GRX-810 at 20 MPa in the vertical direction compared to horizontal GRX-810 also shown in supplementary figure 9. In fact, any as-built AM superalloy would be expected to possess similar debits in creep in the horizontal orientation if they possess similar grain texture⁶⁴.”

3. Page 12: “After 10 cycles, GRX-810 has an oxide layer on the surface of the sample that is 10-20 μm thick, while the 718 sample was oxidized completely through the bulk of the material. 718 also exhibited large porosity and grain growth as shown in 6(a).” It should refer to Figure 5(a).

- This has been fixed in the new version.

4. Page 14: “It is possible that the presence of nano yttria particles in the scale may also assist in adhesion or some other mechanism that assists in oxidation performance similarly to how the inclusion of Y2O3 in the base alloy provides the benefit of inhibiting grain growth in the metal

matrix⁴⁵.” It is necessary to further clarify or modify the statement regarding why "assist in adhesion" is similar to the grain growth inhibition mechanism seen in ODS (oxide dispersion strengthened) metals. Typically, grain growth inhibition is associated with mechanisms related to Zener pinning.

- We agree with Reviewer 2's assessment and have changed that statement as shown below.
 - “It is possible that the presence of nano yttria particles in the scale may also assist in adhesion thus enhancing the oxidation performance of the alloy⁴⁵”
- 5. Page 19: “Figure 7 shows horizontal creep data for two types of GRX-810 specimens: cylindrical specimens printed on the EOS M280 with 25x recycled powder, and flat specimens printed with freshly mixed powder on an EOS M100.” The 25x recycled powder still retains certain properties, it is good. However, for a rigorous comparison, the horizontal creep data from the same powder should be provided.**
- We agreed with this reviewer's assessment and have removed this aspect of study from the paper and plan to perform a more controlled and rigorous study in the near future. We hope reviewer 2 finds the additional creep tests and analysis make the paper much stronger and impactful instead.

Reviewer 3:

This manuscript presents a comprehensive study on the ultra-high temperature stability and mechanical properties of the NASA-developed alloy GRX-810, which is specifically designed for AM applications in extreme environments. The paper outlines a series of high-T tensile, creep, and cyclic oxidation tests conducted at temperatures up to 1316°C, providing valuable insights into the alloy's performance. The authors explore how GRX-810 outperforms conventional superalloys and highlight the benefits of the oxide dispersion strengthening (ODS) mechanism that has been incorporated through a novel powder coating process.

The manuscript is an extension of previous work by the authors (Smith, T. M. et al., Nature 617, 513–518, 2023), which introduced GRX-810. While this paper presents additional data on microstructure, nano oxide formation, and high-temperature performance, it fails to sufficiently clarify the reasons for the alloy's superior performance at high temperatures compared to other alloys. As such, I do not believe the results in this manuscript are impactful enough to justify publication in Nature Communications. A more detailed feedback can be found below:

We hope reviewer 3 will find that the new version of the paper strongly separates itself from the previous study. Below is a list of major changes we have made to the revision to better highlight this fact.

- New tensile results highlighting the difference between build orientations up to 1316°C.
- Horizontal creep performance of GRX-810
- High resolution characterization of oxides in GRX-810 using STEM, STEM-EDS, synchrotron diffraction, DFT, and XRD. This analysis confirms the presence of trigonal Y₂O₃ oxides in GRX-810. This is the first observation of stable trigonal Y₂O₃ found in ambient temperature and pressures.
- Larson Miller parameter and calculated stress exponents for creep in GRX-810 for both vertical and horizontal build orientations.

1. Including page numbers would have made reviewing the manuscript easier.

- We have included page numbers in the newest submission.

2. While the current manuscript presents more data, it seems that many results shown in Figures 3-5 have already been published. These should be clearly indicated as such. In my opinion, the novelty of the results in this manuscript is not significant enough to warrant a separate publication.

- Figure 3 and 4 have been significantly revised to elevate this concern and now include new results that better separate this study from the previous one.

3. The quality of several figures needs improvement. In particular:

a) The font size in Figures 1, 2, and 3 is too small and should be enlarged for readability.

b) The axis labels in Figure 4 are hard to read and should be more prominent.

c) The choice of yellow and light orange colors in some graphs is in my opinion, not ideal, making it difficult to distinguish data points.

d) In Figure 3, the term "Mpa" should be corrected to "MPa."

- The font sizes have been increased in these figures to make them easier to read and understand.

- Figure 4 has been completely replaced in the newest version

- Other than figure 1 this is no longer the case for most of the figures. We feel that the difference in color in figure 1 can be easily discerned but we are open to further changing these colors if reviewer 3 and the editors would like for that to occur.

- All instances of "Mpa" have been corrected to "MPa". Thank you for catching that.

4. It is unclear in what condition (e.g., fine grain (FG) or coarse grain (CG), as-built or HIPped) the samples were tested for cyclic oxidation. Additionally, while the authors mention the excellent oxidation performance of GRX-810, the discussion on this topic is quite superficial. Currently, the focus on cyclic oxidation is overshadowed by the detailed discussion of creep behavior and nano oxide evolution, leaving key insights about the material's long-term oxidation stability unexplored. Expanding on this topic with more in-depth discussion would add considerable value to the manuscript.

- We now note that the GRX-810 tested in oxidation was fine grain and in the as-built state. In addition, we have added the additional oxidation analysis in the paper which we feel better strengthens the discussion.

"To better understand the oxidation results, additional 1300°C cyclic oxidation tests were performed comparing GRX-810 and the current state-of-the-art single crystal blade alloys CMSX-4 and CMSX-10 as shown in Figure 6(a).

Fig. 6: Oxidation and microstructural stability at 1300°C. (a) Additional oxidation results comparing GRX-810 to commonly used single crystal blade alloys at 1300°C. There is an additional GRX-810 test showing the weight change after a 24-hour soak at 1300°C. Metallography cross-sections of (b) superalloy 718 after 7 cycles and (c) GRX-810 after 10 cycles.

The results in Figure 6(a) appear to further confirm the oxidation performance found in Figure 5(c). GRX-810 had a slower weight loss rate compared to CMSX-10. CMSX-4 had the best oxidation performance over the 20 cycles measured; however, after 15 cycles its weight loss rate started to accelerate and exceed that of GRX-810. This change may be caused by a depletion of Al near the surface compromising the alloy's ability to maintain a stable alumina oxide layer. In addition to the cyclic tests, a 24-hour soak at 1300°C was performed on GRX-810 to elucidate the cause of weight loss in the GRX-810 samples at this temperature. The sample was found to gain significant weight during the soak suggesting that the weight loss found in the cyclic tests was dominated by oxide scale spallation upon cooling and not volatilization of chromia at elevated temperatures."

5) The authors should provide a more thorough evaluation of their creep data. Specifically:
 a) Present the minimum creep rates as a function of applied stress for both vertical and horizontal orientations and derive the stress exponent n by plotting the $\log(\text{minimum strain rate})$ vs. $\log(\text{stress})$ and applying Norton's creep law. This analysis would provide insights into the dominant creep mechanism.

b) A microstructural analysis of the creep samples near the fracture surface would also help in understanding the failure mechanisms, especially in relation to the role of nano-dispersoids as well as IM precipitates or elemental segregation on grain boundaries.

- We have included the analysis asked for in the revised paper as shown below and agree with reviewer 3 that it makes the manuscript much more impactful by doing so.

“Orientation effects on GRX-810 creep properties

The creep curves provided in Figures 3 and 4 reveal the influence that grain structure has on the creep performance in GRX-810. An overview of these results with additional creep tests are highlighted in Figure 8.

Fig. 8: Creep Strength Overview of GRX-810. (a) The Larson Miller Parameter vs. stress for creep rupture in GRX-810 in the horizontal and vertical build directions compared to plate Haynes 230. (b) Creep strain rate vs. stress at 1093°C for both orientations and the calculated stress exponents.

The Larson Miller Parameters revealed in Figure 8(a) was calculated using the equation below.

$$Rupture\ LMP = (Temp\ (C^\circ) + 273.15) * (\log t_{hr} + 20) \tag{1}$$

Clear differences in creep rupture life of GRX-810 was observed between the vertical and horizontal build orientations. When designing parts for GRX-810 this creep anisotropy should be considered for complex stress states, however preliminary component demonstrations have not shown unexpectedly poor performance for GRX-810³². Using the same creep tests, the creep stress exponents were calculated from the slope of the minimum strain rate as a function of stress for both orientations as shown in figure 8(b) using the below equation⁵⁷.

$$n = d(\ln(\dot{\epsilon}))/d(\ln\sigma) \tag{2}$$

The stress exponents reveal that the two orientations may be under different creep deformation failure modes. Higher values suggest dislocation creep is dominant while lower values suggest that diffusion creep and grain boundary sliding are more active. The results in Figure 8(b) suggest that grain boundary

sliding mechanisms may be more prominent in horizontal oriented creep compared to the vertical orientation in GRX-810. This result is supported by the fracture surfaces characterized in Supplementary Figure 8. The fracture surface found in the horizontal GRX-810 creep specimen reveals that failure occurs primarily along grain boundaries. In contrast, the vertical GRX-810 creep specimen did not clearly appear to fail along grain boundaries.”

Supplementary Figure 8: Optical microscopy images of creep fracture surfaces in GRX-810 between creep performed in the horizontal and vertical orientations.

6. **The authors mention the well-known phenomenon of creep anisotropy in conventionally manufactured ODS alloys. However, it should be noted that compressive creep anisotropy has also been recently observed in ODS Hastelloy X and ODS Ni-Cr-Al-Ti model alloys, both fabricated via LPBF (<https://doi.org/10.1016/j.addlet.2022.100069>, <https://doi.org/10.1002/adem.202200753>). For the Hastelloy X, the compressive creep stress applied perpendicular to the aligned, textured grains resulted in significantly faster creep in both ODS and non-ODS variants. This suggests that the interplay between texture, grain shape, grain size, and intermetallic grain boundary precipitates is complex, which is likely true for GRX-810 as well. The authors are recommended to incorporate these findings and/or extend the microstructural characterization (see points 7) and 8)) to provide a more nuanced discussion of GRX-810’s anisotropic behavior.**
- We have included in the reference provided by reviewer three in the revised paper and have added additional tests to better understand the creep anisotropy in GRX-810 in both ODS and non-ODS versions as shown below in Supplementary Figure 9. The new results in Figures 3, 4, and supplemental Figure 9, as well as the analysis provided in Figure 8 paints a much clearer picture of how the oxides and grain texture effect the creep performance in GRX-810.

Supplementary Figure 9: As-built horizontal creep rupture lives vs stress for GRX-810 with and without oxides. The diamond marks the creep rupture life of as-built non-ODS GRX-810 in the vertical orientation.

7. **The manuscript would benefit from a more in-depth study of the nano oxides and their impact on the high-temperature performance of GRX-810. As it stands, the discussion does not offer substantial new knowledge beyond the previous publication - even the authors acknowledge that various important questions remain still unanswered. The authors' sub-chapter on nano-oxide characterization focuses on the formation of monoclinic Y₂O₃, probably stabilized by Nb, but does not convincingly explain how this affects the alloy's high-temperature strength or creep performance. The discussion remains speculative, and the manuscript would benefit from stronger evidence linking these observations to the alloy's mechanical properties. More emphasis on how the nano-oxides contribute to strength, oxidation resistance, and creep performance is needed to justify the paper's novelty.**
 - We hope the additional characterization and analysis found in the discussion section on the nano-oxides help improve the paper's novelty. Based on new, extensive analysis, using multiple characterization techniques we confirm for the first time the presence of Type-A trigonal Y₂O₃ nanoparticles in GRX-810. We also reveal that this results in much finer oxide dispersion and thus will improve the high temperature properties of the alloy. We further highlight this by showing new creep tests that compare GRX-810 with and without the oxide dispersion.
8. **In the same chapter, the authors write: "The finding that oxides transform from their initial structure, and their interaction with other elements from the metal matrix, strongly indicate that the oxides melt and reprecipitate during the AM process, which has been hypothesized in recent published work". This statement has already been demonstrated in previous work by other researchers, such as Kenel et al., who observed a similar phenomenon during laser powder bed fusion of an ODS Ni-Cr-Al-**

Ti γ/γ' model alloy. Their study showed the decomposition of Y_2O_3 at the high temperatures during LPBF, leading to Y segregation to grain boundaries and the formation of Y-Ni IM precipitates within grains and on grain boundaries (besides formation $Y_4Al_2O_9$ slag because of the high amount of Al in their alloy (<https://doi.org/10.1016/j.addma.2021.102224>). Given the composition of GRX-810, similar behavior is expected during LPBF processing, and this might affect the creep performance. Thus, I encourage the authors to further investigate the fate of the Y_2O_3 dispersoids during LPBF, the grain boundary properties and their potential influence on creep behavior.

- In all the high-resolution SEM, STEM, EDS, and X-ray diffraction analysis performed on GRX-810 to date we have never observed the presence of Y segregation to the grain boundaries or Y-M intermetallic formation at grain boundaries. I will make note that we have observed this occur in other AM ODS alloys we have tried to print in the past few years but never in GRX-810, which may be one reason for its superior high temperature properties. As this has never been observed in GRX-810, we feel that this topic is out of scope of the present study. I'd be happy to discuss this topic with reviewer 3 in more detail on why Y does not decompose during printing of GRX-810 while it has been observed in other alloys outside this review after this whole process has been completed if they are interested.

Dr. Timothy Smith
NASA Glenn Research Center
21000 Brookpark Rd.
Cleveland Oh, 44145
216-433-2632
timothy.m.smith@nasa.gov

Dear Editors and Reviewers,

We are again excited to have been given the opportunity to revise our manuscript, “The ultra-high temperature stability and properties of GRX-810” for Nature Communications. We have carefully considered the additional comments and concerns made by reviewers 1 & 2 and in the list below describe how each was addressed. We also thank everyone for the patience as we worked to address these concerns. We again have added significantly more analysis and characterization in the revised version of this paper and feel it is a much stronger study as a result. We also hope that the additional time, resources, and characterization we invested in this revision highlights how seriously we considered each of the reviewer’s concerns. We are also greatly appreciative of the effort and time it takes the reviewers to provide their helpful insights. Our responses to each of their comments are provided below.

Reviewer #1 (Remarks to the Author):

The authors have been partially responsive to the reviewers, even adding some additional experiments. However, there are still a number of issues that must be addressed before this paper is publishable.

1. Fig. 1: The authors still have not addressed the softening in the 1093C tensile tests. Were the samples sectioned after testing? Did the creep samples have similar instabilities? Was there recrystallization during tensile testing but not creep testing?

- We have included new analysis that reveals that ODS GRX-810 does not undergo any recrystallization during creep and therefore does not experience similar instabilities as the tensile tests show. The softening is most likely a result of recrystallization. However, we feel additional analysis of these tests is out of scope for the current study. In fact, we have another manuscript that will explore this effect in detail and characterize the recrystallization that occurs during these high temperature tensile tests.

2. Fig. 1: What is the explanation for the 1260C and 1316C samples having essentially the same flow stress in spite of a temperature difference of 56C? What are the serrations in the stress strain curves?

- These tests are for different orientations of AM GRX-810. Therefore, they actual reveal that the horizontal orientation becomes slightly weaker than the vertical orientation as we reach these higher temperatures. You’ll notice that the opposite is true at room temperature. The serrations are noise in the displacement measurement for these tensile tests. We understand that these results are unusual and testing at these temperatures can be difficult so extra scrutiny needs to be made

on these reported values. Since this manuscript has been under review we have been able to confirm these high temperature strengths through multiple outside vendors and internal creep tests. Below is a snapshot of one of the vendors report outs.

ASTM E21 – HIP – 2375°F

Specimen #	Client ID	Diameter in	Modulus Mpsi	0.2% Offset Yield ksi	Ultimate Tensile Stress ksi	Strain at Break %
1	GRX-810 B1-146	0.2505	4.66	5.91	6.55	5.03
2	GRX-810 B1-160	0.2535	6.99	5.96	6.13	4.65
3	GRX-810 B1-169	0.2538	6.79	6.72	6.78	4.47
Mean			6.15	6.19	6.49	4.71
Std. Dev.			1.29	0.45	0.33	0.29

3. One should not have to go to supplementary materials to find the composition of this material. Where the chemical measurements on the powder or the printed material? Are there any differences in oxygen in the fine vs coarse grained material?

We have included the nominal composition of GRX-810 in the introduction section.

“Most notable, was the improved high temperature properties GRX-810 exhibited over other ODS alloys in the same NiCoCr family despite having seemingly similar compositions and microstructures³⁰. One possibility for this difference could be the influence the additional reactive elements in GRX-810’s nominal composition (**Ni-32Co-30Cr-3W-1.5Re-0.8Nb-0.3Ti-0.3Al-0.055C**) may have on the formation of nano-scale Y₂O₃ particles during the AM print process.”

The measured O amount for both powder lots was quite similar and cannot account for any of the property differences between the alloys. FG GRX-810 had 0.019 wt.% O and CG GRX-810 had 0.015 wt.% O according to the vendor certificate of analysis.

4. Fig. 3. How reproducible are the creep results: is the fine grain vs coarse grain effect consistent, given that it is such a small difference in grain size?

- Yes, the creep is quite consistent. We believe the additional test shown in Figure 8 highlights this.

5. If the solidus is only predicted, any claims about strength relative to the solidus should be removed. Is there any DTA data?

- We have included DSC analysis and provided a measured solidus temperature for GRX-810 (1357°C).

Supplementary Figure 1: (a) Electron backscatter diffraction grain orientation maps of as-built Lot 1 GRX-810 produced using an EOS M100 and GRX-810 produced using an EOS M280. The top maps correspond to the XY plane and the bottom maps represent the IPF maps from the orientation map above them. (b) Differential scanning calorimetry (DSC) of FG GRX-810 revealing a solidus of 1357°C

6. The authors still have not added any convincing discussion about the mechanisms of strengthening up to 1316C. The ODS / newly identified carbide strengthening do not seem consistent with the creep stress exponents. Supplementary Fig. 10 (which should have testing conditions listed) should be helpful for this discussion (comparisons of the radii of dislocation loops with applied stress, particle spacing etc) which should appear in the main body.

- We have added lengthy analysis that better reveals the impact the trigonal oxides have on the high temperature properties of GRX-810. The below discussion has been included in the revised manuscript.

“Oxide dispersion strengthening in GRX-810

It remains an open question as to whether this difference in oxide crystal structure can explain some of the high temperature performance differences between the two ODS alloys. In a previous study, Zhou et al.⁵⁷ explored the compositional effect on oxide formation and characteristics for Fe-Cr-Al based ODS alloys. They found that different oxide particles and their crystal structures were correlated with various size distributions of the oxides. In their study, they found monoclinic $Y_4Al_2O_9$ were associated with the finest size distribution. To investigate if a size distribution difference existed between the NiCoCr-ODS (ReB) and GRX-810 alloys, oxide size distributions were extracted

from STEM-EDS Y chemical maps and images as shown in Figure 7(f). Similar to the finding by Zhou et al.⁵⁷, it appears that the trigonal oxides in GRX-810 are significantly finer than the cubic oxides in NiCoCr-ReB. Notably, 90% of the oxides in GRX-810 possessed a diameter less than the average cubic oxides in NiCoCr-ReB ODS alloy. In fact, the average diameter (25nm) of the trigonal oxides in GRX-810 was almost half compared to the average diameter of the cubic oxides found in the NiCoCr-ODS alloy (46nm). This difference, though appearing subtle, suggests significant differences in the oxide size distribution, number density, and particle spacings between the two alloys. Considering the same oxide wt.% were used for both alloys, a volume fraction of oxides can be estimated for each alloy based on the alloy density and the density of the different oxide types. Assuming a uniform distribution of spherical particles, a normal size distribution and using the measured average oxide diameter between the two alloys and calculated volume fractions, the average face-to-face distance between oxide particles (λ) can be determined using the relationship below⁵⁸:

$$\lambda = 1.25\left(\sqrt[3]{1/N}\right) - 2r \quad 1$$

where r is the average radius of the oxide particle. The number density (N) can be determined with the relationship:

$$N = \frac{f}{V_{avg}} \quad 2$$

where f is the volume fraction of the oxide particles and V is the average volume of an oxide particle. Using these relationships the average distance between the particles in GRX-810 and NiCoCr-ReB ODS were determined to be 70nm and 124nm, respectively. Notable is the difference in the calculated number density of oxides between the alloys. Assuming the similar estimated volume fractions between the two alloys, GRX-810 was calculated to have a number density over 5 times greater than NiCoCr-ReB ODS ($1.88 \times 10^{21} \text{ m}^{-3}$ compared to $3.25 \times 10^{20} \text{ m}^{-3}$). Based on the significant oxide differences calculated between the two alloys it should be expected that the high temperature properties between the two alloys will be different. A calculated threshold stress at 1100°C for both alloys can be determined using the relationship below.

$$\sigma_{th} = A \left(\frac{M}{2\pi}\right) \left(\frac{Gb}{\lambda}\right) * \left(\ln\left(\frac{D}{r}\right) + B\right) \quad 3$$

where G is the shear modulus of GRX-810 and NiCoCr-ReB ODS at 1100°C which was derived from E and ν (0.26). M is the Taylor factor (3.0), A and B are constants based on the dislocation characteristics and strengthening mechanisms and are 1.06 and 0.65, respectively. Lastly, D is calculated from the relationship:

$$D = (2r\lambda)/(2r + \lambda) \quad 4$$

Based on these assumptions and relationships, the threshold stress for GRX-810 at 1100°C was calculated to be 32MPa while for NiCoCr-ReB ODS it was less than half that at 15MPa. Though this calculation ignores much of the nuance and microstructure effects that control creep in these alloys (ie, grain size, grain structure, remnant dislocation density, and grain boundary carbides) it clearly reveals the critical influence that particle size can have on the high temperature creep properties of these alloys. The finer Type A trigonal oxides in GRX-810 can account for a 2x increase in the threshold stress compared to the larger type C cubic oxides found in NiCoCr-ReB ODS. As

mentioned previously, earlier studies have explored producing a finer dispersion of oxides in ODS alloys through changes in the oxide chemistry and crystal structure. Most notable, Ti and been shown to promote $Y_2Ti_2O_7$ oxides that form much finer dispersions compared to cubic Y_2O_3 ³⁷. However, this finer dispersion has been found to be a result of a greater coherency between these oxides and the surrounding metal matrix which lowers driving force, thus limiting growth of the oxide⁵⁹. This higher coherency may be detrimental to the oxides ability to pin dislocations, and may promote shearing of the oxides rather than climb-bypass⁶⁰. GRX-810, manufactured using L-PBF, may be unique in that it has promoted a finer dispersion of Y_2O_3 without promoting oxides more coherent with the surrounding matrix, thus maximizing the strengthening potential of the oxide dispersion.

An additional case study was performed for this manuscript to quantify the important contribution that the trigonal Y_2O_3 oxides have on the high temperature creep properties of GRX-810. Two different GRX-810 creep samples, one without the oxides and one with them, were manufactured using the same base powder feedstock, AM printer, and post-processing HIP step. The non-ODS GRX-810 sample was crept at 1093C / 21 MPa, below the threshold stress calculated previously for ODS GRX-810. Both non-ODS and ODS versions were crept under the same conditions for as long as the non-ODS test survived. After the non-ODS GRX-810 sample ruptured, the ODS GRX-810 test was terminated and both samples were characterized. The results are shown below in Figure 8.

Fig. 8: Creep response between ODS and non-ODS GRX-810. (a) The creep curves of non-ODS and ODS GRX-810. The ODS GRX-810 test was terminated after the non-ODS test failed at 16 hours. The deformation and kernal average misorientation maps of (b) ODS GRX-810 and (c) non-ODS GRX-810.

The creep curves reveal vastly different responses between ODS and non-ODS GRX-810 at 1093°C / 21MPa. The non-ODS sample ruptured after 16 hours at 4% strain, while the ODS sample did not creep at all. In fact, the ODS sample appeared to exhibit creep negative during the first few hours of the test. It is not clear if this is a true response or an artifact from the creep frame being revealed due to the lack of strain occurring during the test. Future investigations will further probe this result. Still, the lack of a creep response in the ODS sample further validates the threshold stress calculated in this study. The analysis of the post creep samples highlights the influence that the oxides play in the microstructural stability of GRX-810. In Figure 8(b), the ODS material did not

reveal any deformation, recrystallization, or relaxation after both the HIP and creep test. The kernel average misorientation (KAM) map indicates that the dislocation sub structure present after the AM printing process still remains after creep. In contrast the non-ODS sample revealed a microstructure that fully recrystallized and possessed significant crack formation along grain boundaries. The KAM analysis of the non-ODS material revealed heavy deformation build up at the grain boundaries and subsequent cracks and an absence of the dislocation sub-structure that was initially present after the AM print. This comparison unequivocally confirms the importance these nano-oxides have on the exceptional high temperature properties of GRX-810.”

7. On page 11, the authors claim that the oxidation performance must good since the sample survived tensile testing; however at a rate of 2×10^{-4} /s for 4% strain the test would last approximately 3 min, so this argument should be removed.

- This argument was removed in the revised paper.

8. The authors show no evidence of a Cr₂O₃ film forming at 1300C even in the short term, with only a very low magnification image in Fig. 6. The linear weight loss is a classes sign of vaporization. Unless chromia scales are actually shown and analyzed, these arguments about protective chromia are completely unsupported. The weight loss, given in g/cm², is quite high.

New SEM/EDS characterization of the oxide scale that formed in the 1300C GRX-810 samples has been included in the revised manuscript.

“To better understand the oxidation results, additional 1300°C cyclic oxidation tests were performed comparing GRX-810 and the current state-of-the-art single crystal blade alloys CMSX-4 and CMSX-10 as shown in Supplementary Figure 2. The results in Supplementary Figure 2 appear to further confirm the oxidation performance found in Figure 5(c). GRX-810 had a slower weight loss rate compared to CMSX-10. CMSX-4 had the best oxidation performance over the 20 cycles measured; however, after 15 cycles its weight loss rate started to accelerate and exceed that of GRX-810. This change may be caused by a depletion of Al near the surface compromising the alloy’s ability to maintain a stable alumina oxide layer. In addition to the cyclic tests, a 24-hour soak at 1300°C was performed on GRX-810 to elucidate the cause of weight loss in the GRX-810 samples at this temperature. The sample was found to gain significant weight during the soak suggesting that the weight loss found in the cyclic tests was dominated by oxide scale spallation upon cooling and not volatilization of chromia at elevated temperatures. Figures 6(a) and 6(b) indicate significant differences in microstructural stability between 718 and GRX-810 when cycled at 1300°C.

Fig. 6: Oxidation and microstructural stability at 1300°C. Metallography cross-sections of (a) superalloy 718 after 7 cycles and (b) GRX-810 after 10 cycles. (c) SEM and chemical map revealing a continuous Cr₂O₃ oxide layer with Y₂O₃ in the GRX-810 oxide scale from (b).

Figures 6(a) and 6(b) indicate significant differences in microstructural stability between 718 and GRX-810 when cycled at 1300°C. After 10 cycles, GRX-810 has an oxide layer on the surface of the sample that is 10-20μm thick, while the 718 sample was oxidized completely through the bulk of the material. 718 also exhibited large porosity and grain growth as shown in 6(a). This porosity is most likely caused by incipient melting or rapid diffusion of oxidant and reaction with alloying elements, such as Cr⁴⁸. No such porosity or incipient melting was observed in any of the GRX-810 samples. SEM analysis of the oxide layer formed in GRX-810 at these conditions reveals a continuous Cr₂O₃ oxide layer with Y₂O₃ as shown in Figure 6(c). The observation of a continuous oxide layer, void formation, and Cr depletion near the surface of the sample suggest the oxidation performance of GRX-810 is driven more by spallation than vaporization.”

9. In Fig. 8, the Larson Miller parameter has the wrong units (time units are missing).

- We have added the time units to the Larson miller parameter figure.

10. Testing has been conducted on fine grain, coarse grain, horizontal and vertical with an additional HIP cycle. Every sample and figure should indicate which of these material parameters are being addressed.

The manuscript has been updated to better clarify what material was tested/analyzed for each figure and section.

11. The mechanical testing at 1093C and above is not clearly describe in this section – was it Metcut or AMTT? The gage lengths of the specimens have still not been given.

The below was included in the revised manuscript.

“For the tensile tests at 1093°C and below the gauge length employed was 19.05mm. For the elevated tensile tests above 1093°C the gauge length was 30.4mm. The mechanical testing at 1093°C and below was primarily performed at Metcut Research Inc., with some supplementary tests conducted at the NASA Glenn Research Center using samples that employed a gauge length of 19.05mm.”

12. Supplementary Table 1 – is all data for the same strain rate?

- Yes, and this has been included on the figure and paper.

“Supplementary Table 1: Elevated Tensile data for CG GRX-810. All tests were performed with a 0.305 mm/mm/min strain rate until 4% strain was reached. At 4% elongation the test was transitioned to displacement control at a rate equal to 1.98 mm/mm/min.”

Reviewer #2 (Remarks to the Author):

The reviewer continues to find the discussion on yttrium oxide particles, particularly the potential mechanisms by which the novel crystal structure emphasized by the authors enhances high-temperature performance, insufficiently explored. In the revised manuscript, the authors included a simple statistical analysis of oxide sizes, suggesting that the trigonal structure may result in finer particle sizes compared to conventional oxides. However, this section is inadequately developed and requires expansion. For instance, what is the specific impact of these particles on high-temperature creep behavior? Given that creep in this material is primarily governed by grain boundary sliding, how does the oxide affect this mechanism? Is the substantial performance enhancement solely due to smaller particle sizes? Are there differences in the spatial distribution of yttrium oxide particles compared to other ODS alloys? Additionally, are other factors, such as interfacial effects, contributing to the observed improvements?

Regarding the differentiation of this work from their previous study (Nature, 2023, 617(7961): 513-518.), the authors highlight their discovery and structural characterization of the novel yttrium oxide crystal structure, with additional distinctions limited to higher testing temperatures, different stress conditions, and varying orientations for alloys. The reviewer considers this level of novelty insufficient. Moreover, if the oxide crystal structure is deemed critical to material performance, why is this aspect not mentioned in the abstract or introduction? The authors are strongly encouraged to explicitly emphasize the significance of the new crystal structure in these sections to enhance the manuscript’s coherence and highlight its innovation.

- We agree with reviewer’s 2 comment and concerns over the lack of analysis and discussion pertaining to the trigonal oxides characterized in GRX-810. We have highlighted the significance of their presence throughout the paper and have added a lengthy discussion on the relationship between the trigonal oxides and superior high temperature properties of GRX-810. Below, are the additional analysis and remarks over the oxides in GRX-810 that have been added in the revision.

Abstract:

“The demand for metal alloys that can perform at extreme temperatures above 1100°C while remaining manufacturable has sparked renewed interest in printable oxide dispersion strengthened (ODS) alloys. Recently, NASA developed an ODS alloy designed for additive manufacturing, known as GRX-810, which has demonstrated exceptional tensile and creep performance at temperatures of 1093°C and higher. In the present study, tensile tests of GRX-810 were conducted up to 1316°C and creep tests were performed in both the horizontal and vertical orientations, relative to the build direction. Thermal cycling was executed at 1100°C, 1200°C, and 1300°C in air. The oxidation behavior of GRX-810 is compared to that of alumina forming single crystal Ni-base superalloys and chromia-forming wrought alloys such as superalloys 718 and 625. **High resolution atomic-scale characterization and atomistic modeling are employed to explain the exceptional high temperature properties observed in GRX-810, particularly in relation to the unique, finer trigonal yttrium oxides produced during the additive manufacturing process.**”

Introduction:

“Most notable, was the improved high temperature properties GRX-810 exhibited over other ODS alloys in the same NiCoCr family despite having seemingly similar compositions and microstructures³⁰. One possibility for this difference could be the influence the additional reactive elements in GRX-810’s nominal composition (Ni-32Co-30Cr-3W-1.5Re-0.8Nb-0.3Ti-0.3Al-0.055C) may have on the formation of nano-scale Y_2O_3 particles during the AM print process. Previous studies have demonstrated that Y_2O_3 will acquire other alloying constituents during processing (mechanical alloying, extrusion, etc.) to form modified nano oxides that deviate from a cubic crystal structure typical for Y_2O_3 . Reactive elements such as Al³², Ti³³, Hf³⁴, and Zr³⁵ have all been found to diffuse and react with Y_2O_3 . For example, Al, in some instances, has been observed to react with Y_2O_3 to form monoclinic $Y_4Al_2O_9$ ³⁶. These different oxides have been shown to possess varying particle size, contributing to differences in high temperature strength³⁷.”

“These exceptional results are explained by the presence of the first observed instances of trigonal (space group P-3m1) Y_2O_3 particles that exhibit significantly finer size distributions and greater number densities than the commonly found cubic Y_2O_3 particles characterized in other NiCoCr-based ODS alloys.”

Discussion:

“From Figure 7 and Supplementary Figure 5, it can be concluded that the oxides in GRX-810 are in fact Type A, Trigonal Y_2O_3 nanoparticles. Considering that both NiCoCr-ReB ODS and GRX-810 were manufactured using the same process and the same initial Y_2O_3 nanoparticles, the difference in oxide crystal structures between the two alloys suggests that alloy composition, in combination with the rapid solidification of AM, can promote the appearance of Type-A trigonal Y_2O_3 nanoparticles. While, fully describing the formation of these unique oxides will be the scope of future studies, some initial considerations are provided here. *Ab initio* molecular dynamics (AIMD) simulations reveal that the trigonal phase is mechanically stable only at very high (close to melting) temperatures. Thus, its appearance in combination with the small sizes of the Y_2O_3 particles in GRX-

810 is most likely associated with melting and precipitation of Y_2O_3 during AM⁵⁵. This is broadly consistent with experiments, which show a phase transition from the cubic phase to the hexagonal H-type phase around 100 degrees Celsius below the melting temperature⁵⁶. The reactive elements in GRX-810 may help promote both the H-type and/or A type phases, as is suggested in Figure 7e. The promotion and rapid quenching of the H-type phase could result in formation of the Type-A trigonal phase, which then remains due to the rapid solidification of the metal surrounding the oxide as a consequence of the large difference between the trigonal and cubic phase, as shown in supplementary Figure 8. This may explain why the trigonal phase is promoted and observed in the GRX-810 at the room temperature though it is not stable in bulk at these conditions.”

“Oxide dispersion strengthening in GRX-810

It remains an open question as to whether this difference in oxide crystal structure can explain some of the high temperature performance differences between the two ODS alloys. In a previous study, Zhou et al.⁵⁷ explored the compositional effect on oxide formation and characteristics for Fe-Cr-Al based ODS alloys. They found that different oxide particles and their crystal structures were correlated with various size distributions of the oxides. In their study, they found monoclinic $Y_4Al_2O_9$ were associated with the finest size distribution. To investigate if a size distribution difference existed between the NiCoCr-ODS (ReB) and GRX-810 alloys, oxide size distributions were extracted from STEM-EDS Y chemical maps and images as shown in Figure 7(f). Similar to the finding by Zhou et al.⁵⁷, it appears that the trigonal oxides in GRX-810 are significantly finer than the cubic oxides in NiCoCr-ReB. Notably, 90% of the oxides in GRX-810 possessed a diameter less than the average cubic oxides in NiCoCr-ReB ODS alloy. In fact, the average diameter (25nm) of the trigonal oxides in GRX-810 was almost half compared to the average diameter of the cubic oxides found in the NiCoCr-ODS alloy (46nm). This difference, though appearing subtle, suggests significant differences in the oxide size distribution, number density, and particle spacings between the two alloys. Considering the same oxide wt.% were used for both alloys, a volume fraction of oxides can be estimated for each alloy based on the alloy density and the density of the different oxide types. Assuming a uniform distribution of spherical particles, a normal size distribution and using the measured average oxide diameter between the two alloys and calculated volume fractions, the average face-to-face distance between oxide particles (λ) can be determined using the relationship below⁵⁸:

$$\lambda = 1.25(\sqrt[3]{1/N}) - 2r \quad 1$$

where r is the average radius of the oxide particle. The number density (N) can be determined with the relationship:

$$N = \frac{f}{V_{avg}} \quad 2$$

where f is the volume fraction of the oxide particles and V is the average volume of an oxide particle. Using these relationships the average distance between the particles in GRX-810 and NiCoCr-ReB ODS were determined to be 70nm and 124nm, respectively. Notable is the difference in the calculated number density of oxides between the alloys. Assuming the similar estimated volume fractions between the two alloys, GRX-810 was calculated to have a number density over 5 times greater than NiCoCr-ReB ODS ($1.88 \times 10^{21} \text{ m}^{-3}$ compared to $3.25 \times 10^{20} \text{ m}^{-3}$). Based on the

significant oxide differences calculated between the two alloys it should be expected that the high temperature properties between the two alloys will be different. A calculated threshold stress at 1100°C for both alloys can be determined using the relationship below.

$$\sigma_{th} = A \left(\frac{M}{2\pi} \right) \left(\frac{Gb}{\lambda} \right) * \left(\ln \left(\frac{D}{r} \right) + B \right) \quad 3$$

where G is the shear modulus of GRX-810 and NiCoCr-ReB ODS at 1100°C which was derived from E and ν (0.26). M is the Taylor factor (3.0), A and B are constants based on the dislocation characteristics and strengthening mechanisms and are 1.06 and 0.65, respectively. Lastly, D is calculated from the relationship:

$$D = (2r\lambda)/(2r + \lambda) \quad 4$$

Based on these assumptions and relationships, the threshold stress for GRX-810 at 1100°C was calculated to be 32MPa while for NiCoCr-ReB ODS it was less than half that at 15MPa. Though this calculation ignores much of the nuance and microstructure effects that control creep in these alloys (ie, grain size, grain structure, remnant dislocation density, and grain boundary carbides) it clearly reveals the critical influence that particle size can have on the high temperature creep properties of these alloys. The finer Type A trigonal oxides in GRX-810 can account for a 2x increase in the threshold stress compared to the larger type C cubic oxides found in NiCoCr-ReB ODS. As mentioned previously, earlier studies have explored producing a finer dispersion of oxides in ODS alloys through changes in the oxide chemistry and crystal structure. Most notable, Ti and been shown to promote $Y_2Ti_2O_7$ oxides that form much finer dispersions compared to cubic Y_2O_3 ³⁷. However, this finer dispersion has been found to be a result of a greater coherency between these oxides and the surrounding metal matrix which lowers driving force, thus limiting growth of the oxide⁵⁹. This higher coherency may be detrimental to the oxides ability to pin dislocations, and may promote shearing of the oxides rather than climb-bypass⁶⁰. GRX-810, manufactured using L-PBF, may be unique in that it has promoted a finer dispersion of Y_2O_3 without promoting oxides more coherent with the surrounding matrix, thus maximizing the strengthening potential of the oxide dispersion.

An additional case study was performed for this manuscript to quantify the important contribution that the trigonal Y_2O_3 oxides have on the high temperature creep properties of GRX-810. Two different GRX-810 creep samples, one without the oxides and one with them, were manufactured using the same base powder feedstock, AM printer, and post-processing HIP step. The non-ODS GRX-810 sample was crept at 1093C / 21 MPa, below the threshold stress calculated previously for ODS GRX-810. Both non-ODS and ODS versions were crept under the same conditions for as long as the non-ODS test survived. After the non-ODS GRX-810 sample ruptured, the ODS GRX-810 test was terminated and both samples were characterized. The results are shown below in Figure 8.

Fig. 8: Creep response between ODS and non-ODS GRX-810. (a) The creep curves of non-ODS and ODS GRX-810. The ODS GRX-810 test was terminated after the non-ODS test failed at 16 hours. The deformation and kernel average misorientation maps of (b) ODS GRX-810 and (c) non-ODS GRX-810.

The creep curves reveal vastly different responses between ODS and non-ODS GRX-810 at 1093°C / 21MPa. The non-ODS sample ruptured after 16 hours at 4% strain, while the ODS sample did not creep at all. In fact, the ODS sample appeared to exhibit creep negative during the first few hours of the test. It is not clear if this is a true response or an artifact from the creep frame being revealed due to the lack of strain occurring during the test. Future investigations will further probe this result. Still, the lack of a creep response in the ODS sample further validates the threshold stress calculated in this study. The analysis of the post creep samples highlights the influence that the oxides play in the microstructural stability of GRX-810. In Figure 8(b), the ODS material did not reveal any deformation, recrystallization, or relaxation after both the HIP and creep test. The kernel average misorientation (KAM) map indicates that the dislocation sub structure present after the AM printing process still remains after creep. In contrast the non-ODS sample revealed a microstructure that fully recrystallized and possessed significant crack formation along grain boundaries. The KAM analysis of the non-ODS material revealed heavy deformation build up at the grain boundaries and subsequent cracks and an absence of the dislocation sub-structure that was initially present after the AM print. This comparison unequivocally confirms the importance these nano-oxides have on the exceptional high temperature properties of GRX-810.”

“The presence of trigonal oxides in GRX-810 is notable as it is the first observation of trigonal Y_2O_3 stable at ambient pressure and temperatures. This crystal structure appears correlated to a finer oxide dispersion present in GRX-810, resulting in a greater oxide number density, and reduced inter-particle distance compared to ODS alloys that possess cubic Y_2O_3 particles. Ultimately, this unique oxide formation in GRX-810 results in significantly improved high temperature properties.”

Dr. Timothy Smith
NASA Glenn Research Center
21000 Brookpark Rd.
Cleveland Oh, 44145
216-433-2632
timothy.m.smith@nasa.gov

Dear Editors and Reviewer 1,

We are again grateful to have been given the opportunity to revise our manuscript, “The ultra-high temperature stability and properties of GRX-810” for Nature Communications. We have carefully considered the additional comments and concerns made by reviewer 1 and in the list below describe how each was addressed.

Reviewer #1 (Remarks to the Author):

Reviewer #1 (Remarks to the Author):

This manuscript has been substantially improved, with the additional material on oxide characterization, strengthening calculations and other details of materials and testing added. Nearly all reviewer concerns have been addressed. The only area not adequately addressed relates to oxidation.

In the text, the authors refer to “one-hour isothermal cycling”. Since it refers to a cyclic oxidation experiment, this should be re-worded to report heating and cooling rates and the fact that one hour at temperature was employed. The GRX 810 loses weight (on the g/cm³ scale) at all temperatures – 1100, 1200 and 1300 °C. The authors state that: “Superalloys 625 and GRX-810 completed the full 200 1-hr cycles and remained intact without extensive weight loss or distortion.” The reviewer suggests that the “without extensive weight loss” is not justified. Comparing to B1900 at 1200 °C, is not really helpful since this alloy was never designed to operate near that temperature. The presence of a chromia layer following cooling is used to conclude “oxide stability near melting”. This layer was likely strongly influenced by the cooling cycle. If the authors want to claim that chromia is thermodynamically stable at 1300 °C, then they should justify this with thermodynamic calculations.

- We thank Reviewer #1 for their diligence and helpful comments for providing clarity to the oxidation experiments. We have added more detailed information regarding the heating/cooling rates in the “Oxidation Testing” portion of the “Methods” section. Additionally, we agree that further context was necessary regarding the stability and protective nature (or lack thereof) of the chromia scale that forms. In the “Elevated temperature cyclic oxidation” section we have added a number of comments and removed some phrases to make it clear that although the GRX-810 maintains its shape and does not undergo catastrophic failure, the Cr₂O₃ layer that forms is not considered a protective scale due to the consistent weight loss that is observed at all temperatures. We have also added an additional reference on the sublimation rates of Cr₂O₃ to provide further context on the mass loss rates due to evaporation and oxide spallation with respect to the observed weight

loss. Again, we sincerely appreciate the reviewer's comments and hope that these changes are sufficient to address the concerns.

Below are the additional/edited descriptions that were added in the manuscript to address these concerns.

- "Chromia forming alloys, ME3 (a disk alloy) and 718 failed catastrophically (sample weight below 1 gram) at 60 and 80 hours, respectively. Superalloys 625 and GRX-810 completed the full 200 1-hr cycles and remained intact without distortion. Ultimately, both traditional and ODS chromia forming alloys exhibited more weight loss than the alumina forming single crystal B-1900 alloy up to 1200°C." – pg. 14
- "The samples maintained their integrity as well, showing minimal distortion after 10 hours of exposure (inset in Figure 5(c)) although there was observable thinning of the sample." – pg. 15
- Experiments performed by Wang *et al.* on the sublimation of Cr₂O₃ at high temperatures projected $\sim 5 \times 10^{-8}$ gm/cm²-sec⁴⁸. The weight loss observed in Figure 5(c) was on the order of $\sim 2 \times 10^{-4}$ gm/cm²-sec, so it is assumed that the weight change was dominated by oxide scale spallation and not volatilization of chromia at elevated temperatures. Additionally, a 24-hour soak at 1300°C was performed on GRX-810 caused significant weight gain, which further supports the hypothesis that the weight loss in cyclic testing is from spallation and not sublimation." – pg. 16
- "Cooling cycles between 1-hour cycles occurred hanging in lab air for 20 min. Samples were inserted into the hot zone at a rate of ~ 25 mm/s, reached 90% of the target temperature within 90s, and were comfortably at the target temperature within 5 min. For the cooling cycle, the samples were removed at a rate of ~ 25 mm/s, were below 500°C within ~ 60 s and ~ 50 °C within 10 min." pg. 35.

Lastly, we included the B-1900 alloy in the oxidation study because we believed it could help illustrate to readers the superior oxidation properties that alumina-forming Ni-base superalloys possess over chromia-forming alloys, even if they have not been optimized for that temperature environment or specific property. We feel that the data presented showcases this well. As has been mentioned in previous responses, GRX-810 was not developed to replace alumina forming single crystal blade alloys. However, by presenting the properties of alumina-forming single crystal and conventional wrought superalloys, we believe this helps the reader better understand the capabilities and limitations of GRX-810 as a high temperature alloy.